# OC-CLIP : Object-centric Binding in Contrastive Language-Image Pre-training

## Abstract

Recent advancements in vision-language models (VLMs) have been driven by contrastive models like CLIP (Radford et al., 2021), which learn to associate visual information with their corresponding text descriptions. However, these models have limitations in understanding complex compositional scenes involving multiple objects and their spatial relationships. To address these challenges, we propose a novel approach that diverges from commonly used strategies relying on the design of hard-negative augmentations. Our work instead focuses on integrating sufficient inductive biases into pre-trained CLIP-like models to improve their compositional understanding without using additional data annotations. To that end, we introduce a binding module that connects a scene graph, derived from a text description, with a slot-structured image representation, facilitating a structured similarity assessment between the two modalities. We also leverage relationships as text-conditioned visual constraints, thereby capturing the intricate interactions between objects and their contextual relationships more effectively. Our resulting model (OC-CLIP) not only enhances the performance of CLIP in multi-object compositional understanding but also paves the way towards more accurate and efficient image-text matching of complex scenes.

## 1 Introduction

Recent advancements in multi-modal representation learning have primarily been enabled by the introduction of CLIP (Radford et al., 2021). CLIP learns aligned image-text representations from Internet-scale data. Despite its success, CLIP exhibits limitations in understanding complex scenes composed of multiple objects (Kamath et al., 2023; Yuksekgonul et al., 2023a; Doveh et al., 2023; Paiss et al., 2023). For instance, while capable of recognizing individual objects, CLIP struggles with interpreting spatial relationships among objects in the scene (*e.g.*, "the cat is to the left of the mat" *vs.* "the cat is to the right of the mat") and adequately associating objects with their corresponding attributes (*e.g.*, "a red square and a blue circle" *vs.* "a blue square and a red circle"). The process of acquiring this compositional understanding of the world is known as the *binding problem* in the literature, and may be decomposed into *segregation*, *representation*, and *composition* problems (Greff et al., 2020b).

Efforts to improve the compositional understanding of CLIP-like models have largely relied on leveraging *hard negative examples*, either in the text space (Kalantidis et al., 2020; Yuksekgonul et al., 2023b; Zhang et al., 2024b; Doveh et al., 2023; Paiss et al., 2023) – to improve sensitivity to the order of words and subtle textual differences – or the image space (Awal et al., 2024; Le et al., 2023; Zhang et al., 2024a) – to improve sensitivity to subtle visual differences. Although these methods have somewhat improved CLIP-like models' performance on scene compositionality benchmarks (Parcalabescu et al., 2022; Zhao et al., 2022; Yuksekgonul et al., 2023b; Hsieh et al., 2023b), they do not explicitly address the binding problem as they focus mainly on enhancing the model's representation capabilities with additional data, hindering their generalization to unseen scene compositions.

Yet, the object-centric representation learning literature (Eslami et al., 2016; Greff et al., 2020a; Locatello et al., 2020; Wu et al., 2023; Seitzer et al., 2023) has long focused on developing methods to address the segregation and representation problems as a way to facilitate the subsequent compositional processing of images. This has led to the development of inductive biases to segregate

different objects in a scene into distinct representational *slots*, which have been shown to naturally scale to an increasing number of visual objects and relations (Locatello et al., 2020; Webb et al., 2023; Mondal et al., 2024; Elsayed et al., 2022). To the best of our knowledge, advances in object-centric representation learning are yet to be explored in the vision-language domain.

Therefore, in this paper, we focus on enhancing the compositional scene understanding of CLIP-like models by leveraging the advances from object-centric representation learning. In particular, we propose to endow CLIP-based vision-language architectures with segregation and composition capabilities. Our core idea is to adapt the slot-centric representation paradigm for CLIP architectures and dynamically align each representational slot with the object entities mentioned in the text. To do so, we design a binding module that connects a scene graph, derived from the textual description, with a slot-structured image representation. We utilize the scene graph's relationships as constraints to effectively capture the complex interactions among the visual entities represented as slots. Our enhanced model, which we refer to as Object-Centric CLIP (OC-CLIP), not only boosts CLIP's performance in understanding multi-object compositional scenes but also improves the accuracy of image-text matching in complex and highly compositional visual scenarios.

Our contributions are summarized as follows:

- We introduce OC-CLIP, a model which endows CLIP-based architectures with segregation and composition capabilities, effectively addressing the binding problem.
- We evaluate the sample efficiency of our approach against methods leveraging hard negative augmentations in a controlled 3D environment and show the overall efficiency of OC-CLIP compared to both text and image based a hard-negative augmentations.
- We demonstrate that OC-CLIP significantly enhances the binding of object-centric attributes and spatial relationships across a representative set of challenging real-world compositional image-text matching benchmarks. Notably we report an increase of +**16.1%** accuracy in the challenging *swap attribute* split of SugarCrepe compared to OpenCLIP (Ilharco et al., 2021) finetuned in-domain and go from random chance to more than **93%** on COCO-spatial and **95%** GQA-spatial from the Whatsup benchmark (Kamath et al., 2023).
- We show the scaling potential of OC-CLIP when trained from scratch on a noisy CC12M (Changpinyo et al., 2021) dataset.

## 2 RELATED WORK

**Contrastive Pretraining of VLMs.** Vision-language models (VLMs) have made substantial strides in both the vision and multi-modal domains (Bordes et al., 2024). Modern VLMs are pretrained on vast, diverse and oftentimes noisy multi-modal datasets (Changpinyo et al., 2021; Schuhmann et al., 2022; Ilharco et al., 2021; Zeng et al., 2022a) and applied to various zero-shot tasks. CLIP (Radford et al., 2021) presented a contrastive learning approach used for pretraining, which involves training the model to differentiate between similar and dissimilar image-text pairs. This approach encourages the model to learn a shared representation space for images and text, where semantically similar pairs are close together and dissimilar pairs are far apart. Following CLIP's lead, image-text contrastive learning has become a prevalent strategy for VLM pretraining (Liu et al., 2023; Cai et al., 2024; Liu et al., 2024a; Dai et al., 2023; Zhai et al., 2022b; Chen et al., 2022; Beyer et al., 2024; Fini et al., 2023). Contrastive vision-language pretraining spans numerous downstream applications, including zero-shot image classification (Zhai et al., 2022a; Radford et al., 2021; Metzen et al., 2024; Gao et al., 2021), text-to-image generation (Podell et al., 2023; Abdal et al., 2021; Ramesh et al., 2022; Saharia et al., 2022), as well as assessing text-image alignment (Moens et al., 2021; Cho et al., 2023). In this work we are particularly interested the ability of CLIP-based VLMs to evaluate compositional text-image alignment.

**Compositional Understanding Benchmarks.** Several benchmarks have been developed to assess the compositional understanding of VLMs. In this work, we focus on benchmarks structured as cross-modal retrieval tasks where the model needs to distinguish between correct and incorrect text descriptions given an image, and evaluations are based on accuracy metrics. The majority of these benchmarks (Zhao et al., 2022; Yuksekgonul et al., 2023a; Parcalabescu et al., 2022) rely on the rule-based construction of negative captions and the generation of their associated image counter-factuals (Zhang et al., 2024a; Awal et al., 2024). Yet, many of these benchmarks

may be solved by leveraging the language prior exclusively (Goyal et al., 2017; Lin et al., 2024), hence disregarding the information from the visual input. To address this, benchmarks such as SugarCrepe (Hsieh et al., 2023a) leverage large language models to generate plausible and linguistically correct hard negatives, and show that previously introduced text-based hard negative strategies are not always effective (Yuksekgonul et al., 2023b) – *e.g.*, when considering attribute and object swaps between textual descriptions. Other benchmarks focus on assessing the VLMs' spatial understanding (Kamath et al., 2023; Yuksekgonul et al., 2023b; Zhang et al., 2024a), and propose to finetune CLIP-based models on data containing a high proportion of spatial relationships since these relationships tend to underrepresented in commonly used pretraining datasets. Interestingly, Kamath et al. (2023) show that even when finetuning with in-domain data with an overrepresentation of spatial relationships, state-of-the-art models still exhibit a close to random chance performance. In this work, we test the hypothesis that spatial relationship failures are due to the lack composition in the similarity score computation used to train CLIP-like models.

**Object-centric Binding Inductive Biases.** CLIP has been shown (Yuksekgonul et al., 2023a) to be pushed to learn disentangled, bag-of-words-style representations from the contrastive loss and the easily distinguishable negatives typically used for pretraining. Although the learned representations might be effective for objects presented in isolation, they struggle with scenes containing multiple objects (Tang et al., 2023). For example, consider a simple scene with a green apple and a yellow banana. In this case, the model must maintain and correctly link the attributes ("green", "yellow") to the objects ("apple", "banana"), without mixing the concepts – *e.g.*, "yellow apple" or 'green banana". This exemplifies the importance of devising robust mechanisms within the CLIP architecture and/or training to accurately handle multiple objects, while preventing feature interferences. In this work, we focus on equipping CLIP with object-centric binding inductive biases and take inspiration from the architectures proposed in the unsupervised object-centric visual representation learning literature (Locatello et al., 2020; Wu et al., 2023; Seitzer et al., 2023; Assouel et al., 2022). Many recent image-only approaches follow a simple inductive bias introduced by slot Attention (Locatello et al., 2020), where an image – encoded as a set of input tokens – is soft partitioned into K slots. In particular, attention maps are computed via an **inverted cross attention** mechanism (Wu et al.), where the softmax is applied along the query dimension in order to induce a competition between the slots to explain different groups of input tokens. In this work, we extend these inductive biases to define text-conditioned visual slots from the input image.

## 3 METHOD

Our goal is to enhance CLIP-based architectures with segregation and composition capabilities. Our method starts by extracting representations of distinct objects and relationships in a textual description, as well as representations of patches in an image. Next, a binding module matches the text representation of objects to the relevant image patches, producing a slot-centric representation of the image. Finally, a structured similarity score compares the slot-centric representation with the textual representations of different objects, and leverages the extracted relationships as constraints applied to the visual slots. Our key contributions lie in the design of the binding module[1] and the proposal of the structured similarity score, which we detail below. Figure 1 presents an overview of the proposed approach.

**Notation.** We denote as $\mathbf{x}$ an image of shape $\mathbb{R}^{h \times w \times 3}$ and as $\bar{\mathbf{x}} = [\bar{\mathbf{x}}^1, ..., \bar{\mathbf{x}}^N] = E_\phi(\mathbf{x}) \in \mathbb{R}^{N \times d}$ its patch-level encoding, where $E_\phi$ is an image encoder – typically a pre-trained ViT (Dosovitskiy et al., 2020) – $N$ is the number of patches and $d$ the dimensionality of the patch embeddings. We denote as $t$ the text description, or caption, associated with $\mathbf{x}$. We extract a scene graph, $\mathcal{G}$ from $t$ by leveraging an LLM-based parsing approach. $\mathcal{G}$ is composed of a set of nodes $\mathcal{N} = \{N^1, ..., N^M\}$ representing the $M$ objects in $t$ and of a set of edges $\mathcal{E} = \{(\mathbf{r}^1, s^1, o^1), ..., (\mathbf{r}^P, s^P, o^P)\}$ representing the $P$ relationships in $t$. Each relationship is represented by a tuple $(\mathbf{r}, s, o)$, where $\mathbf{r}$ is the embedding of the predicate, $s$ the subject and $o$ the object of the relationship. For example, the scene graph of "A red apple to the left of a blue car" will be represented with the set of nodes {"red apple", "blue car"} and the set of edges {("to the left of", "red apple", "blue car")}. In practice, we represent $\mathcal{N}$ as a matrix of node features $\mathbf{N}$, where each row contains the embedding of a node in the graph. Moreover, we represent each $s^i$ and $o^i$ in the relationship tuples as indices referencing the nodes (rows) in $\mathbf{N}$.

---

[1]Code for the Binding Module is given in the Appendix 9

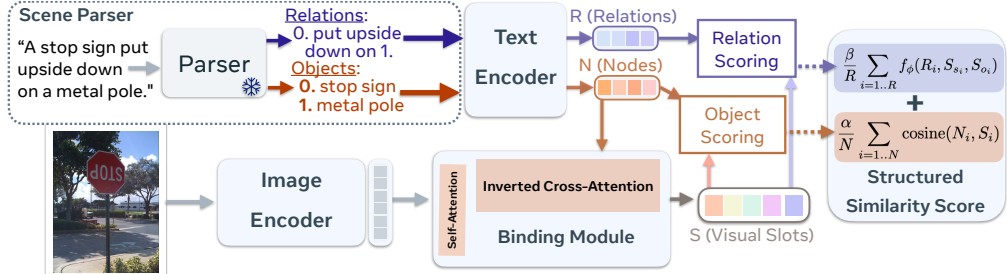

Figure 1: Object-Centric CLIP (OC-CLIP) overview. OC-CLIP begins with scene parsing, where we utilize a text parser (*e.g.*, Llama3-based) to extract objects and relations from the input caption. The extracted text objects and relations are then fed into a text encoder, which generates distinct text embeddings for both entities and relations. In parallel, the corresponding image is processed by an image encoder to produce patch-level image embeddings. These image embeddings are then combined with the text entity embeddings and passed through a *binding module*, which outputs visual token slots embeddings. To align the text entity embeddings with the visual token slots, we use an *object scoring* function that learns to map the text entities to their corresponding slots. Furthermore, we introduce a *relation scoring* function that encourages the visual slots to incorporate relationship information, thereby enriching the representation.

## 3.1 BINDING MODULE

Our first contribution resides in the binding module. The idea is that when comparing the content of a caption and an image we do not want the features of different objects to interfere with each other but rather keep them separate at a representational level. The role of the binding module is thus to extract a slot-centric representation of an image where the content of the slots are pushed to represent the nodes of the associated scene graph.

To do so, we implement the binding module using a *inverted* cross-attention layer (Wu et al.), where the queries are the nodes from our scene graph and the keys and values are the image patches. We normalize the attention coefficients over the queries' dimension in order to introduce a competition between queries to explain different parts of the visual input. We follow common practice and set the attention's softmax temperature to $\sqrt{D}$, with $D$ being the dimensionality of the dot-product operation. Applying the softmax along the queries' dimension pushes all the candidate keys to be softly matched to at least one query. However, captions mostly describe specific parts of the image, and rarely capture all the visual information. Since we want only the relevant visual information to be captured by the queries, we add a set of default query tokens, stored in a matrix $\mathbf{Q}_{\text{default}}$, which participate in the competitive attention mechanism – with the goal of absorbing the visual information not captured in the caption. These default query tokens are dropped in the subsequent computation steps of our model (akin to registers in ViT backbones (Darcet et al., 2024)). We find the default query tokens crucial to stabilize the training our model.

The binding module computations are formalized as follows:

$$\mathbf{Q} = \mathbf{W}_q \mathbf{N},$$
$$\mathbf{K}, \mathbf{V} = \mathbf{W}_k \bar{\mathbf{x}}, \mathbf{W}_v \bar{\mathbf{x}},$$
$$\mathbf{Q}' = [\mathbf{Q}; \mathbf{Q}_{\text{default}}],$$
$$\text{Attention}(\mathbf{Q}', \mathbf{K}, \mathbf{V}) = \text{softmax}\left(\frac{\mathbf{Q}' \cdot \mathbf{K}^T}{\sqrt{D}}, \text{dim='queries'}\right) \cdot \mathbf{V},$$
$$\mathbf{S}, \mathbf{S}_{\text{default}} = \text{Attention}(\mathbf{Q}', \mathbf{K}, \mathbf{V}). \tag{1}$$

Here, $\mathbf{W}_q$, $\mathbf{W}_k$, and $\mathbf{W}_v$ are the linear projection weight matrices for the queries, keys, and values, respectively, $\mathbf{S}$ are the visual slots, $\mathbf{S}_{\text{default}}$ are the visual slots from default query tokens, which are discarded for subsequent steps, and [.] denotes the concatenation operation.

Thus, the output of this binding module are the visual slots $\mathbf{S}$. Intuitively, these slots are pushed to represent the visual objects, or entities, that correspond to the nodes of the scene graph. Their object-centric learning is driven by the structured similarity that we detail in the next section.

## 3.2 STRUCTURED SIMILARITY SCORE

Our second contribution resides in the introduction of a structured similarity score, whose goal is to promote the constraints imposed by the scene graph on the learnable visual slots. Our proposed structured similarity score is composed of an *object scoring* function and a *relationship scoring* function. The object scoring function assesses the presence of each node in the scene graph (objects present in the caption). We model this function as the sum of the cosine similarity between each textual node representation $\mathbf{N}^i$ and its assigned visual slot $\mathbf{S}^i$. The relationship scoring function encourages the relational constraints imposed by each edge in the scene graph and is defined as a learnable function $f_\phi$ of the relationship embedding $\mathbf{r}^i$, and the visual slot representations $\mathbf{S}^{s^i}$ and $\mathbf{S}^{o^i}$ corresponding to the subject and object of the relationship, respectively. We derive the overall structured similarity score over the visual slots $\mathbf{S}$ from an image $\mathbf{x}$ and a graph $\mathcal{G} = (\{N^i\}_{i=1..M}, \{(\mathbf{r}^i, s^i, o^i)\}_{i=1..P})$ such that:

$$S(\mathbf{x}, \mathcal{G}) = \frac{\alpha \sum_{i=1..M} \text{cosine}(\mathbf{N}^i, \mathbf{S}^i) + \beta \sum_{i=1..P} f_\phi(\mathbf{r}^i, \mathbf{S}^{s^i}, \mathbf{S}^{o^i})}{\alpha M + \beta P}, \tag{2}$$

where $\alpha$ and $\beta$ are learned parameters controlling the strength of each score. $M$ and $P$ are the number of nodes and relationships in the scene graph $\mathcal{G}$, respectively.

We define $f_\phi$ as follows:

$$f_\phi(\mathbf{r}, \mathbf{S}^s, \mathbf{S}^o) = \text{cosine}(\mathbf{r}, f_s([\mathbf{r}, \mathbf{S}^s]) + f_o([\mathbf{r}, \mathbf{S}^o])), \tag{3}$$

where [.] denotes the concatenation of two vectors and $f_s$ and $f_o$ are MLPs that reduce the dimensionality of their inputs. Note that we model the relationship scoring function so that it keeps the same scale as the object scoring function and can take the order of the relationship into account.

## 3.3 TRAINING

The model is trained using the following loss:

$$\mathcal{L} = \mathcal{L}_{itc} + \mathcal{L}_{rel}. \tag{4}$$

$\mathcal{L}_{itc}$ is the image-text contrastive loss defined to minimize the distance between image and scene graph representations from paired text-image data while maximizing the distance between image and scene graph representations from unpaired text-image data as:

$$\mathcal{L}_{itc} = -\sum_{i=1}^{B} \left( \log \frac{\exp^{S(\mathbf{x}_i, \mathcal{G}_i)}}{\sum_{j=1}^{B} \exp^{S(\mathbf{x}_j, \mathcal{G}_i)}} + \log \frac{\exp^{S(\mathbf{x}_i, \mathcal{G}_i)}}{\sum_{j=1}^{B} \exp^{S(\mathbf{x}_i, \mathcal{G}_j)}} \right), \tag{5}$$

where $B$ is the number of elements in the batch. Note that the $S$ is the structured similarity score defined in Eq. 2. $\mathcal{L}_{rel}$ is the loss that pushes the model to learn a non-symmetric relationship scores:

$$\mathcal{L}_{rel} = -\sum_{i=1}^{B} \log \frac{\exp^{S(\mathbf{x}_i, \mathcal{G}_i)}}{\exp^{S(\mathbf{x}_i, \mathcal{G}_i)} + \exp^{S(\mathbf{x}_i, \bar{\mathcal{G}}_i)} + \exp^{S(\mathbf{x}_i, \tilde{\mathcal{G}}_i)}}, \tag{6}$$

where $\bar{\mathcal{G}}$ and $\tilde{\mathcal{G}}$ are altered scene graphs. In $\bar{\mathcal{G}}$, we swap the order of the subject and the object of a relationship, whereas in $\tilde{\mathcal{G}}$, we randomly chose the relationship's subject and object from the nodes in the scene graph. We ablate the main components of OC-CLIP in Table 6 and give a more extensive ablation analysis in Apppendix A.1

## 4 RESULTS

We evaluate OC-CLIP in two different setups. In the first setup, we leverage synthetic data to control for the combinations of objects and relationships seen during training. We demonstrate that, unlike vanilla CLIP with OpenCLIP weights (Ilharco et al., 2021), OC-CLIP generalizes well to combinations of objects and attributes not seen during training. In the second setup, we utilize real-world datasets and benchmarks to further evaluate OC-CLIP, and highlight that our model can also improve performances *w.r.t.* the OpenCLIP baseline.

## 4.1 COMPOSITIONAL UNDERSTANDING IN A CONTROLLED 3D ENVIRONMENT (PUG)

In this section, we study the object-centric binding problem, and the sample efficiency of hard-negative-based baselines against our proposed OC-CLIP. We consider a controlled 3D environment based on PUG (Bordes et al., 2023), where the vocabulary is fixed and where the models are exposed to every *object-attribute* conjunction. We build a dataset composed of a single textured animal, or pairs of animals, in different backgrounds. We use a combination of 4 textures, 20 animal classes, and 5 different backgrounds, see example in Figure 2a and 7a. We test the compositionality of learned representations along several generalization axes. The evaluation is based on image-text retrieval tasks where we assess both attribute binding understanding and spatial relational understanding (in Appendix section A.5.1). We follow prior benchmarks (Hsieh et al., 2023a) and perform text-retrieval only between the correct caption and the associated negative caption.

The goal of this initial experiment is to determine if OC-CLIP can effectively separate objects from their attributes. To do so, we test the model's ability to generalize to object-texture combinations not seen during training. We create splits with varying proportions of animal pairs to be used for training, and a held out a test set of unseen object-attribute pairs combinations. We ensure that each pair of animals is assigned a unique set of attributes during training. For instance, if a tortoise and an elephant appear together in the training set, they will only appear as "red tortoise" and "blue elephant." We consider two generalization axes: (1) **Seen object pairs** This axis tests for unseen object-attribute combination of animal pairs seen during training. (2) **Unseen object pairs:** This axis tests for unseen pairs of animals, regardless of the attributes associated to them.

We finetune both the CLIP architecture with OpenCLIP weights (Ilharco et al., 2021) – hereinafter referred to as OpenCLIP – and OC-CLIP on data splits containing an increasing proportion of the data from the PUG environment. We consider increasing the number of seen animal pairs and design the hard negatives required to train the models as images containing pairs of animals in the training set but with swapped attributes. We then test both models on image-text retrieval tasks and report the results in Figure 2. Figure 2(b) shows the results for the seen object pairs generalization, whereas Figure 2(c) presents the results for the unseen object pairs. As shown in the figure, when we do not have any hard negative and only use a low number of animal pairs for training, the baseline OpenCLIP model shows poor performance across both generalization tasks, whereas our OC-CLIP is able to generalize well. In particular, OC-CLIP reaches 100% accuracy on seen object pairs regardless of the amount of hard negatives and object pairs shown during training, which translates into an absolute 28% increase over the OpenCLIP baseline in the most challenging case. For unseen object pairs, OC-CLIP exhibits consistent improvements over OpenCLIP as well, *e.g.* over 20% absolute improvement in the most challenging case. These results highlight the better sample efficiency of OC-CLIP, even when using a high proportion of hard negatives in the training set.

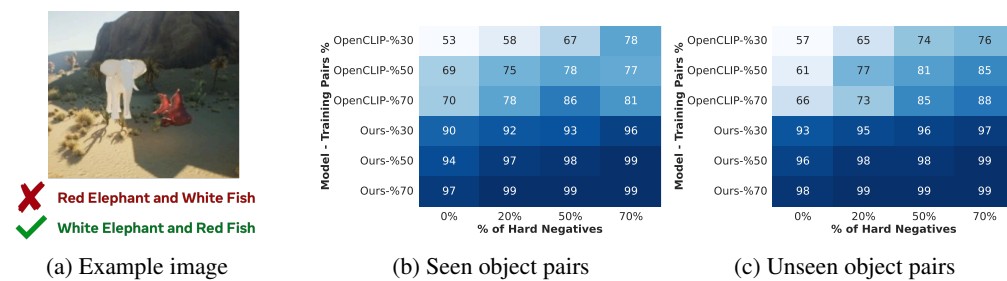

(a) Example image    (b) Seen object pairs    (c) Unseen object pairs

Figure 2: **Attribute Binding.** Performance of the finetuned OpenCLIP and OC-CLIP models on a binary classification task between a caption and its corresponding hard-negative, as shown in Figure (a). To assess the models' performance, we compute the accuracy across two dimensions. The first one is the percentage of animal pairs (y-axis) seen during training (animals like elephants and fish could be seen either alone or with other animals but never together). The second dimension (x-axis) is the number of hard-negatives used in the training data. For instance, whether we have the combination "red elephant" and "white fish" in the training data while we only have "white elephant" and "red fish" in the test data.

## 4.2 COMPOSITIONAL UNDERSTANDING IN REAL WORLD DATASETS

In this section, we verify that the observations made in the controlled environment presented in Section 4.1 also transfer to real-word datasets.

**Datasets.** We train OC-CLIP and finetune OpenCLIP in-domain on a set of datasets relevant for real-world compositional understanding. The training text descriptions representing positive samples are taken from COCO (Lin et al., 2014), Visual-Genome (Krishna et al., 2017) and GQA (Hudson and Manning, 2019). The latter annotates images coming from Visual Genome (Krishna et al., 2017) with objects and both spatial and non-spatial relationships, and thus contains a high representation of spatial prepositions. We evaluate the different models on the most challenging benchmarks representative of compositional understanding, ensuring that we validate both their *attribute binding* and *spatial relationship* understanding capabilities. In particular, we use SugarCrepe (Hsieh et al., 2023b) and ARO-A (Yuksekgonul et al., 2023a) for attribute binding and ARO-Relation (ARO-R) (Yuksekgonul et al., 2023a), COCO-spatial and GQA-spatial (Kamath et al., 2023) for spatial relationship understanding. We also include evaluations on Winoground (Thrush et al., 2022) and VL-Checklist (Zhao et al., 2023) in Table 5 and further detail the datasets in the appendix.

**Training.** The training of the OC-CLIP's binding module is done from scratch along with the finetuning of the text and vision backbones. The text backbone is initialized from OpenCLIP weights (Ilharco et al., 2021). We consider 2 different image base ViT backbones, OpenCLIP (ViT-B-16) (Ilharco et al., 2021) and Dinov2 (ViT-B-14) (Oquab et al., 2024), to show the flexibility of our binding module and learned structured similarity score. We use a batch size of 128 and a learning rate of $2e^{-4}$ to train OC-CLIP for 100 epochs. We use a batch size of 256 – following previous finetuning approaches (Kamath et al., 2023; Yuksekgonul et al., 2023b) – and a learning rate of 4e-6 for 20 epochs to finetune the OpenCLIP baseline. We run all the models for 3 seeds and report the mean performance along with their standard deviation.

**Baselines.** We report the performance of a representative set of strong baselines which we separate in two groups: the first group of baselines are VLMs trained contrastively and finetuned in-domain (on COCO) and the second group are hard-negative-based methods. For the first group, we include OpenCLIP – referred to as OpenCLIP-FT –, BLIP (Li et al., 2023a), and XVLM (Zeng et al., 2022a). BLIP is augmented an image-text matching loss and XVLM uses bounding boxes to assist the object-centric binding. Note that these two baselines are also equipped with a language modeling objective which may help identify unplausible captions. For the second group, we select a representative set of hard-negative-based methods to compare to. These methods augment the dataset with rule-based text hard-negatives (NegCLIP (Yuksekgonul et al., 2023b)), language-model-based hard-negatives (CE-CLIP Zhang et al. (2020)), and image-&-language-model-based hard-negatives (CLIP-CC (Zhang et al., 2024a)).

### 4.2.1 ATTRIBUTE BINDING EVALUATION

We evaluate the attribute binding capabilities of OC-CLIP and baselines on SugarCrepe (Hsieh et al., 2023b) and ARO-A (Yuksekgonul et al., 2023b) benchmarks. We report the results in Table 1. When comparing OpenCLIP-FT to OC-CLIP (both models), we observe notable performance boosts on ARO-A and SugarCrepe's swap-attribute, and swap-object. In particular, OC-CLIP B-14 shows a performance boost of +22.1% on ARO-A, whereas in SugarCrepe, our model reaches improvements of +16.1% on the swap-attribute split, +17.7% on the swap-object split, and a smaller +4.7% on the replace-relationship split. Moreover, both OC-CLIP models perform similarly to OpenCLIP-FT on the remaining SugarCrepe splits. This is to be expected since the remaining splits do not require precise binding to distinguish between positive and negative captions and may therefore be solved with a bag-of-words-like representation. When comparing with additional contrastive-based models (BLIP and XVLM) finetuned with in-domain data, both OC-CLIP models show notable improvements on SugarCrepe's swap splits – *e.g.*, OC-CLIP B-14 results in +14.6% in object-swap and +12.3% in attribute-swap – despite not relying on additional binding annotations, nor language modeling losses. The results of BLIP and XVLM on ARO-A may be explained by the use of their use of a language modeling prior; Hsieh et al. (2023a) emphasizes that language-only models are performing well on this benchmark because the negative caption are often not realistic. Both OC-CLIP models also improve the results of hard-negative-based methods on SugarCrepe's swap splits as well as ARO-A. In all the remaining splits of SugarCrepe, except add-attribute,

OC-CLIP models perform similarly to previous works leveraging hard-negatives. The results achieved by CE-CLIP and CC-CLIP on the add-attribute split could be attributed to an increase of attribute coverage induced by the language model generations.

| Model | Swap | | Add | | Replace | | |
|---|---|---|---|---|---|---|---|
| | Object | Attribute | Object | Attribute | Object | Attribute | Relation |
| *Zero-shot* | | | | | | | |
| OpenCLIP | 68.2 | 66.2 | 82.7 | 80.3 | 93.8 | 82.8 | 67.3 |
| *In-domain ft baselines* | | | | | | | |
| BLIP Li et al., 2022b† | 66.2 | 76.2 | - | - | **96.5** | 81.9 | 68.35 |
| XVLM Zeng et al. 2022b † | 64.9 | 73.9 | - | - | 95.2 | 87.7 | 77.4 |
| OpenCLIP-FT | 63.1 $\pm0.6$ | 72.4$\pm1.1$ | **93.4** $\pm0.2$ | 83.1 $\pm0.5$ | 95.4 | 87.0 $\pm0.6$ | 75.5 $\pm0.6$ |
| *Hard-Negative - small scale* | | | | | | | |
| NegCLIP Yuksekgonul et al. (2023a)† | 75.2 | 75.4 | 88.8 | 82.8 | 92.7 | 85.9 | 76.5 |
| CE-CLIP Zhang et al. (2024b)† | 72.8 | 77 | 92.4 | 93.4 | 93.1 | 88.8 | 79 |
| CC-CLIP Zhang et al. (2024a)† | 68.6 | 73.6 | 86.7 | 90.3 | 95.9 | 87.9 | 76.2 |
| *Hard-Negative/Dense Captioning - large scale* | | | | | | | |
| DAC-LLM | 75.1 | 74.1 | 89.7 | **97.7** | 94.4 | **89.3** | **84.4** |
| DAC-SAM | 71.8 | 75.3 | 87.5 | 95.5 | 91.2 | 85.9 | 83.9 |
| *Ours* | | | | | | | |
| OC-CLIP B-16 | 76.3 $\pm0.7$ | 87.1 $\pm0.2$ | 91.3 | 83.8 $\pm1.0$ | 93.9 $\pm0.4$ | 88.3 $\pm0.1$ | 77.0 $\pm0.2$ |
| OC-CLIP B-14 | **80.8** $\pm0.7$ | **88.5** $\pm0.4$ | 93.0$\pm0.3$ | 83.8 $\pm1.1$ | 95.7 $\pm0.4$ | 88.8 $\pm0.6$ | 80.2 $\pm0.2$ |

Table 1: Performance on SugarCrepe. Both OpenCLIP-FT and OC-CLIP are initialized with the same OpenCLIP checkpoints. OC-CLIP is trained with two ViT base backbones with different resolutions: OpenCLIP's backbone (B-16) and Dinov2 (B-14).

### 4.2.2 RELATIONSHIP UNDERSTANDING EVALUATION

We evaluate the spatial relationship understanding capabilities of OC-CLIP and baselines on COCO-spatial, GQA-spatial, and ARO-Relation (ARO-R). Note that ARO-Relation contains both spatial and non-spatial relations but about half of the test examples consists of left/right relationships understanding. We report the results in Table 2 and Table 5 and show consistent improvements of both OC-CLIP models over the baseline models and across the 3 datasets. In particular, the best OC-CLIP model outperforms OpenCLIP-FT by +47.9% on COCO-spatial, +46.6% on GQA-spatial, and +34.7% on ARO-R. When compared to contrastive VLMs finetuned with in-domain

| Model | COCO-spatial | GQA-spatial |
|---|---|---|
| XVLM | 73.6 | 67 |
| BLIP | 56.4 | 52.6 |
| NegCLIP | 46.4 | 46.7 |
| OpenCLIP-FT | 45.6 $\pm0.2$ | 49.1$\pm1.1$ |
| OC-CLIP (B-16) | 90.1 | 93.9 |
| OC-CLIP (B-14) | **93.5** | **95.6** |

Table 2: **Spatial relationship understanding: Performance on COCO-spatial, GQA-spatial from the Whats'up Benchmark** We finetune both OpenCLIP (OpenCLIP-FT here) and OC-CLIP in-domain on COCO, Visual Genome, and GQA data. Both models are initialized with the same OpenCLIP checkpoints.

data (XVLM, BLIP), OC-CLIP models exhibit superior performance, with improvements between +10% and +27% over the strongest contrastive finetuned VLM. Finally, when compared to baselines leveraging hard-negatives (NegCLIP), OC-CLIP remains the highest performer.

### 4.3 GENERALIZATION

In the previous sections, we tested OC-CLIP on datasets that are in-distribution w.r.t. the finetuning on COCO – Note that SugarCrepe is based on COCO images. Here, we test the compositionality performance of OC-CLIP on data distributions different than the one used for model fine-tuning. Specif-

ically, we test the generalization of our model on the challenging Winoground benchmark (Thrush et al., 2022). In Winoground, each sample consists of two image-text pairs, where both texts in the sample present small differences resulting from object, relationship, or object&relationship swaps with their corresponding image. The task involves two types of retrieval tasks: text-based retrieval and image-based retrieval as described in (Thrush et al., 2022). We report the results in Table 3. We observe that both OC-CLIP models consistently outperform OpenCLIP-FT and NegCLIP across all tasks (text, image, group) by a significant margin: +(7.3, 5.1, 2.2)% and +(6.3, 4.0, 2.6)% with B-14 and B-16 backbones, respectively. We also remark that the overall low absolute scores can be partially attributed to the very challelling nature of Winoground, which have been shown to contain some ambiguous/unsolvable pairs, as well as pairs that to be solved require very high image resolution (much higher than 224 to which we operate), see Diwan et al. (2022).

### 4.3.1 Does OC-CLIP work on noisy data?

In order to show the potential of OC-CLIP to learn from scene-graph obtained from a non human-curated captioning dataset we train both ViT-B-16 OpenCLIP model and OC-CLIP from scratch on CC12M (Changpinyo et al., 2021). We did not tune the hyperparameters and used the same hyperparameters as suggested in (Mu et al., 2021). Both models are trained for 20 epochs, using a batch size of 4096, a learning rate of $5e - 4$, 1k steps learning rate warmup and a cosine decay after. As recommended by

| Model | Winoground | | |
| | Text Score | Image Score | Group Score |
|---|---|---|---|
| OpenCLIP-FT (COCO) | 25.6 | 11.5 | 7.8 |
| NegCLIP | 29.5 | 10.5 | 8.0 |
| OC-CLIP$_{\text{B-16 (COCO)}}$ | 36.8 $_{\pm 3.1}$ | 14.5$_{\pm 0.6}$ | 10.6 $_{\pm 1.5}$ |
| OC-CLIP$_{\text{B-14 (COCO)}}$ | **37.8** $_{\pm 1.1}$ | **15.6**$_{\pm 1.7}$ | **10.2** $_{\pm 1.1}$ |

Table 3: **Results on generalization.** Winoground is evaluated with the text, image and group scores introduced in Thrush et al. (2022).

Mu et al. (2021) we used AdamW optimizer with 0.5 of weight decay and $\beta_2$ set to 0.98. Interestingly, in Table 4, OC-CLIP shows performance gains in general zero-shot classification (+9.2% in ImageNet) while maintaining a significant gap in zero-shot compositional understanding with a notable +15.9% and +14.3% in the swap attribute and swap object SugarCrepe splits, respectively. This experiment shows that the structured training of OC-CLIP is also effective when scaling to an automatic alt-text captioned dataset and does not only rely on high-quality human captions. We additionally report extensive zero-shot downstream classification performance on the ELEVATER (Li et al., 2022a) suite in Appendix Table 8 and leave further scaling for future work.

| Model | Zero-Shot Classification | | | | | | Compositional | |
| | Food101 | CIFAR10 | CIFAR100 | Eurosat | STL10 | ImageNet | Swap Obj | Swap Att |
|---|---|---|---|---|---|---|---|---|
| CLIP | 36.8 | 55.7 | 28.9 | 26.9 | 87.4 | 29.7 | 60.4 | 61.5 |
| OC-CLIP | 51.0 | 74.3 | 41.5 | 16.9 | 89.8 | 39.5 | 74.7 | 77.4 |

Table 4: Comparison of CLIP and OC-CLIP models on zero-shot classification and compositional understanding tasks. Both are trained from scratch on CC12M for 20 epochs with a batch size of 4096. ViT-B-16. Extensive results from ELEVATER benchmark (Li et al., 2022a) in Table 8

### 4.4 Ablations

In Table 6 we ablate the key design choice of our model and further discuss them in Appendix A.1. Specifically we investigated two key components of the model: the use of competitive cross attention and the local graph contrastive loss. The results showed that removing the competitive cross attention mechanism had a slight impact on fine-grained attribute binding performance, but not on relational understanding. On the other hand, removing the local graph contrastive loss significantly impacted downstream relational understanding, with accuracy decreasing from 80.7 to 73.1 for swap obj and from 80.6 to 74.7 for replace rel. These findings highlight the importance of the local graph contrastive loss in improving the model's relational understanding capabilities.

## 5 Conclusion and limitations

**Conclusion.** In this paper, we proposed Object-Centric CLIP (OC-CLIP), a method to enhance the compositional scene understanding of CLIP-like models by leveraging advances from object-centric

| Model | VL-Checklist | | | ARO | | | |
|---|---|---|---|---|---|---|---|
| | Object | Relation | Attribute | Attribution | Relation | COCO-order | Flickr-order |
| CLIP | 80.0 | 63.0 | 67.4 | 63.2 | 60.0 | 47.9 | 60.2 |
| BLIP | 82.2 | 70.5 | 75.2 | 63.2 | 60.0 | 47.9 | 60.2 |
| XVLM | 85.8 | 70.4 | 75.1 | 73.4 | 86.8 | - | - |
| *Hard-negative Methods* | | | | | | | |
| CLIP-SVLC | 85.0 | 68.9.7 | 72.0 | 73.0 | 80.6 | 84.7 | 91.7 |
| NegCLIP | 84.1 | 63.5 | 70.9 | 71 | 81 | 86 | 91 |
| CE-CLIP | 84.6 | 71.8 | 72.6 | 76.4 | 83.0 | - | - |
| *Dense captioning+Hard-Negative* | | | | | | | |
| DAC-LLM$_{500k}$ | 66.5 | 56.8 | 57.4 | 63.8 | 60.1 | 50.2 | 61.6 |
| DAC-LLM$_{3M}$ | 87.3 | 86.4 | 77.3 | 73.9 | 81.3 | 94.5 | 95.7 |
| DAC-SAM$_{3M}$ | 88.5 | 89.7 | 75.8 | 70.5 | 77.2 | 91.2 | 93.9 |
| DCI | 80.7 | 70.1 | 68.7 | 67.6 | 76.2 | 88.6 | 91.3 |
| DCI$_{neg}$ | 88.4 | 61.3 | 70.4 | 62.0 | 57.3 | 39.4 | 44.6 |
| OC-CLIP | 84.5 | 73.9 | 73.7 | 82.0 | 86.5 | 94.2 | 84.8 |

Table 5: Results (%) on VL-Checklist and ARO Benchmark.

| Local Loss | Competitive X-Attn | Default Token | Relation Module | Swap Obj | Swap Att | Replace Att | Replace Rel |
|---|---|---|---|---|---|---|---|
| ✓ | ✓ | 4 | Additive | 80.7 | 88.7 | 88.3 | 80.6 |
| - | ✓ | 4 | Additive | 73.1 | 88.3 | 89.2 | 74.7 |
| ✓ | - | 0 | Additive | 80.4 | 86.0 | 86.2 | 80.6 |
| ✓ | ✓ | 1 | Additive | 79.9 | 88.4 | 86.7 | 80.9 |
| ✓ | ✓ | 4 | MLP | 78.4 | 87.8 | 87.1 | 78.7 |

Table 6: Ablation of OC-CLIP components. Fine-grained accuracy on SugarCrepe splits.

representation learning. Our approach adapts the slot-centric representation paradigm to CLIP and dynamically aligns each representational slot with the objects mentioned in the text description. This is achieved by the introduction of a binding module and a structured similarity score that allows to train OC-CLIP in a contrastive way. We evaluated the sample efficiency of our approach against common hard-negative augmentation strategies in a controlled 3D environment and showed the overall efficiency of OC-CLIP compared to both text and image-based hard-negative augmentations. We also demonstrated that OC-CLIP significantly enhances the binding of object-centric attributes and spatial relationships across a representative set of challenging real-world compositional image-text matching benchmarks. Notably, we reported an increase of +16.1% accuracy in the challenging swap-attribute split of SugarCrepe compared to OpenCLIP finetuned with in-domain data and drastically improved performance on COCO-spatial and GQA-spatial from the Whatsup benchmark, moving from random chance to more than 93%. Finally we show the scaling potential of OC-CLIP to be trained from scratch on a noisy CC12M (Changpinyo et al., 2021) datastet. Notably we report performance gain in zero-shot classfication (+9.2% in ImageNet 8) while maintaining a significant gap in zero-shot SugarCrepe swap attribute (+15.9%) and swap obj (+14.3%) splits.

**Limitations.** Our current implementation builds upon existing pre-trained backbones and only trains the binding and scoring modules from scratch. This allows us to leverage the knowledge captured by these pre-trained backbones while still adapting to the specific task of compositional scene understanding. Future work could explore ways to improve the scalability of our approach, such as developing more efficient training methods or exploring alternative architectures with similar object-centric inductive biases. We also expect the capacity needed for the text encoder to be reduced since it does not need to encode whole scene configuration but rather single objects and relationships as shown in our CC12M experiemnts and further explained in Appendix A.2.

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

# A APPENDIX

## A.1 ABLATIONS

In this section we ablate and discuss some important design choice of OC-CLIP. We separately ablate and discuss :

- The **similarity score** coefficients $\alpha$ and $\beta$ that control the weight of the objects and relations in the global graph-image similarity score.
- **Binding module inductive biases** and their impact on compositional understanding performance.
- **Local Loss** impact on downstream compositional understanding of relationships.

Important ablation results are summarized in Table 7 and further commented below.

| Model | Loc Loss | Comp Att | Default Token | Relation Module |
|---|---|---|---|---|
| OC-CLIP | ✓ | ✓ | 4 | Additive |
| - Loc Loss | - | ✓ | 4 | Additive |
| - Comp Att | ✓ | - | 0 | Additive |
| + Default Token (1) | ✓ | ✓ | 1 | Additive |
| + MLP Relation | ✓ | ✓ | 4 | MLP |

| Split | Swap Obj | Swap Att | Replace Att | Replace Rel |
|---|---|---|---|---|
| Baseline | 80.7 | 88.7 | 88.3 | 80.6 |
| - Loc Loss | 73.1 | 88.3 | 89.2 | 74.7 |
| - Comp Att | 80.4 | 86.0 | 86.2 | 80.6 |
| + Default Token (1) | 79.9 | 88.4 | 86.7 | 80.9 |
| + MLP Relation | 78.4 | 87.8 | 87.1 | 78.7 |

Table 7: Ablation Experiments, Fine-grained accuracy (% performance on representatitve Sugar-Crepe splits.

**Similarity Score** OC-CLIP's structured global similarity score is a combination of the object and relationship components respectively weighted by two learnt parameters $\alpha$ and $\beta$ balancing the different contributions. We let the model learn those parameters throughout the training. However, during preliminary experiments we tested a different combinations of initial coefficient within the [1.5, 1, 0.5, 0.1] grid and noticed that the model was always converging to a $\frac{\alpha}{\beta} \sim 3$ without any difference in the downstream compositional performance. We thus fix the initial coefficients to $\alpha = 1.5$ and $\beta = 0.5$ and treat them as parameters.

**Default Token and Competitive Cross Attention** In the binding module we propose to use an inductive biases to encourage the query tokens to attend to different groups of patches. In order to do so we use a competitive attention mechanism, the so called inverted cross attention common to many object-centric image encoder architecture (Locatello et al., 2020; Wu et al.). We found that the use of inverted cross attention impacts slightly the fine-grained attribute binning performance (see swap att and replace att performance in Table 7, -Comp Att model does not use any inverted cross attention and is rather implemented with a regular cross attention mechanism, the softmax being done along the keys dimensions.). Interestingly the fine-grained attribute splits only seem to be affected by this design choice and not the splits related to relational understanding.

**Local Graph Contrastive Loss** In designing the structured similarity score of OC-CLIP the relational component is formulated as the following cosine similarity $f_\phi(\mathbf{r}, \mathbf{S}^s, \mathbf{S}^o) = \text{cosine}(\mathbf{r}, f_s([\mathbf{r}, \mathbf{S}^s]) + f_o([\mathbf{r}, \mathbf{S}^o]))$. In theory both $f_s([\mathbf{r}, \mathbf{S}^s])$ and $f_o([\mathbf{r}, \mathbf{S}^o])$ can collapse to ignore the subject object visual representation. In order to prevent such collapse we propose to add a local graph contrastive loss that shares similarity with hard-negative based learning. We enforce

the model to model with a higher similarity the graph composed of the same nodes but with either swapped object and subject indices or shuffle objects and subjects indices within the local graph. In both of those cases the relation component of the structured similarity score becomes (for a single relation graph) :

$$\text{swap } \tilde{G}; \cosine(\mathbf{r}, f_s([\mathbf{r}, \mathbf{S}^s]) + f_o([\mathbf{r}, \mathbf{S}^o]) \tag{7}$$

$$\text{swap } \tilde{G}; \cosine(\mathbf{r}, f_s([\mathbf{r}, \mathbf{S}^o]) + f_o([\mathbf{r}, \mathbf{S}^s]) \tag{8}$$

$$\text{shuffle } \bar{G}; \cosine(\mathbf{r}, f_s([\mathbf{r}, \mathbf{S}^{j!=s}]) + f_o([\mathbf{r}, \mathbf{S}^{i!=o}]) \tag{9}$$

This prevents the model from collapsing because ground-truth $G$ is distinguishable from $\tilde{G}$ and $\bar{G}$ only if the visual representations are not ignored in the relationships components. We ablate incorporating both of those perturbed graphs in Figure 3 and removing the local loss from the training objective in Table 7. Removing the local loss effectively impacts downstream relational understanding on SugarCrepe with a swap obj accuracy decreasing from $80.7$ to $73.1$ and a replace rel accuracy decreasing from $80.6$ to $74.7$ showing the effectiveness of the local graph contrastive loss.

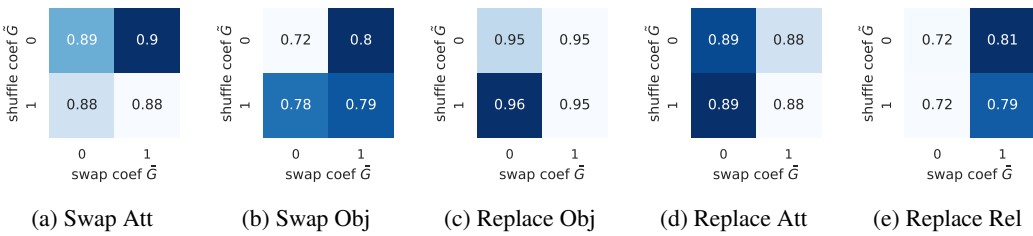

(a) Swap Att  (b) Swap Obj  (c) Replace Obj  (d) Replace Att  (e) Replace Rel

Figure 3: **Local Perturbed Graphs Ablation** In this ablations we keep the initialization seed fixed and include a perturbed graph as a negative sample inside the loss by swapping the order of the subject and object (y-axis), $\bar{G}$ or sampling random subject and object within the positive scene graph (x-axis), $\tilde{G}$.

**Scoring dimensionality**  Our structured similarity score allows the text encoder to focus on encoding information about individual objects and their relationships, rather than the entire scene configuration. To achieve this, we experimented with different dimensionality for both the object scoring bottleneck and the relationship scoring bottleneck. Specifically, each of these scores is designed as a cosine distance between a text representation and a visual component (as described in Section 3.2), with each operating at a bottleneck dimension of $d_{\text{obj}}$ and $d_{\text{rel}}$. In contrast, OpenCLIP represents both the scene caption and the visual representation at a dimension of $d = 512$. We expect that our model can operate effectively at a much lower dimensionality, as it requires less capacity to encode single objects and relationships. We present an ablation study of these two dimensions in Figure 4.

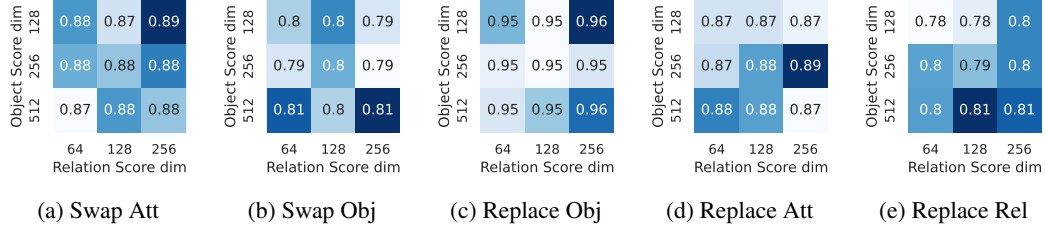

(a) Swap Att  (b) Swap Obj  (c) Replace Obj  (d) Replace Att  (e) Replace Rel

Figure 4: **Score dimensionality ablations** In this ablations we keep the initialization seed fixed and vary the dimensionality of the relation score $d_{\text{rel}}$ (x-axis) and object score $d_{\text{obj}}$(y-axis) and report the performance on the swap and replace splits of sugarcrepe.

## A.2  SCALING EXPERIMENTS.

In the compositional understanding experiments we compare our approach with data-centric finetuning methods that do not add any additional parameters. These methods are expected to retain some

| | Food-101 | CIFAR-10 | CIFAR-100 | SUN397 | Cars | Aircraft | DTD | Pets | FER-2013 | STL-10 | EuroSAT | RESISC45 | GTSRB | KITTI | Country211 | PCAM | UCF101 Frames | CLEVR | HatefulMemes | MNIST | SST2 | ImageNet |
|---|---|---|---|---|---|---|---|---|---|---|---|---|---|---|---|---|---|---|---|---|---|---|
| CLIP | 36.8 | 55.7 | 28.9 | 41.7 | 14.9 | 2.8 | 15.9 | 50.8 | 3.7 | 87.4 | 26.9 | 22.9 | 7.5 | 36.2 | 3.5 | 59.6 | 33.9 | 13.1 | 49.8 | 13.3 | 46.7 | 29.7 |
| OC-CLIP | 51.0 | 74.3 | 41.5 | 49.7 | 18.9 | 3.1 | 18.5 | 65.6 | 8.2 | 89.79 | 16.8 | 35.6 | 13.0 | 33.8 | 5.6 | 53.1 | 41.7 | 15.7 | 47.0 | 13.1 | 49.8 | 39.5 |

Table 8: Zero-shot evaluation of CLIP vs OC-CLIP. Trained on CC12M for 20 epochs.

of the general capabilities of the initial backbone. In contrast, our binding and relationship modules is trained from scratch, which means it may not generalize as well to unseen data and can only be expected to work well within the vocabulary domain it has been exposed to (eg. COCO/VG/GQA in our experiments setting). However an interesting question would be to asses whether such inductive biases and structured similarity object might have some sclaing potential on noisy and non human curated datasets such as CC12M (Changpinyo et al., 2021). To answer that question we propose to train both CLIP and OC-CLIP architectures from scratch on CC12M and compare both of their general understanding and compositional downstream performance. In addition to the zero-shot evaluation, we also provide a computational analysis of the binding module to gain insights into its behavior and limitations.

**Training Details** In order to show the potential of OC-CLIP to learn from scene-graph obtained from a non human-curated captioning dataset we train both ViT-B-16 OpenCLIP model and OC-CLIP from scratch on CC12M (Changpinyo et al., 2021). We did not tune the hyperparameters and used the same hyperparameters as suggested in (Mu et al., 2021). Both models are trained for 20 epochs, using a batch size of 4096, a learning rate of $5e-4$, 1k steps learning rate warmup and a cosine decay after. As recommended by Mu et al. (2021) we used AdamW optimizer with 0.5 of weight decay and $\beta_2$ set to 0.98.We report extensive zero-shot downstreeam classification performance on the ELEVATER (Li et al., 2022a) suite in Table 8. OC-CLIP shows performance gains in both zero-shot classification (+10% in ImageNet) and this experiments show that structured training of OC-CLIP can scale to automatic alt-text captioning dataset. We leave further scaling for future work as the main focus of our work is to emphasize the binding problem that arises when using a vector-based representation and a set of inductive biases as a way of operating on a more structured representation (eg. scene graph).

**Computational analysis of OC-CLIP** In OC-CLIP the visual and text modalities representations are no longer independent (as opposed to CLIP). A image representation is the results of some text-conditioned mechanism operated by the binding module. It essentially extracts relevant visual slots that constitutes the nodes of the scene graph coming from the caption. As a result, there is some notable computational overhead introduced by the additional cross-attention operations of the binding module. In particular :

- 1. The text encoder needs to encode the $N$ nodes and $R$ relations of the scene graph as opposed to a single sentence encoding in CLIP.
- 2. For each Image-Graph pair, The $N$ text nodes cross-attends to $N_{im}$ patches of the ViT in order to extract the structured visual slots.

When training OC-CLIP from scratch we propose to mitigate those two overheads respectively by :

- 1. Using a smaller embedding width (256 vs 512) and number of layers (6 vs 12) in the text encoder. Indeed OC-CLIP only need to encode information about objects and relationships and we expect such encoding to require much less capacity than an encoder that needs to encode a whole caption composed of multiple objects and relations between them.
- 2. We operate on a reduced embedding space 256 for the binding module and thus first project the ViT-B-16 patches from a 768 to a 256 embedding space before computing the nodes to patch cross attention logits.

We only perform experiments with a B-16 architecture for the ViT but perform the computational analysis fro both B and L backbones. We report the results in Table 9 We note that there is a

significant overhead with a base architecture 2.2x but since the binding module perform the same number of operations no matter what the ViT is we show that when scaling the ViT backbone, the binding module is not the bottleneck anymore and the computational overhead is reduced (1.3x).

| Model | ViT Backbone | Text (w,l,ctx) | Binding Module GFLOPs | Text GFLOPS | Vision GFLOPs | Total GFLOPs |
|-------|--------------|----------------|-----------------------|-------------|---------------|--------------|
| OC-CLIP | B | (256, 6, 20) | 12(*num workers) | 180 | 1k | 2.2x |
| CLIP | B | (512, 12, 77) | - | 186 | 1k | 1x |
| OC-CLIP | L | (256, 6, 20) | 12(*num workers) | 180 | 4.9k | 1x |
| CLIP | L | (512, 12, 77) | - | 186 | 4.9k | 1.3x |

Table 9: Computational Comparison of CLIP and OC-CLIP. Calculations are made for a local batch size (per GPU) of 64. We give the Total GFLOPs based on a global batch size of 8192 (=128 num workers). When scaling the ViT backbone the computational overhead of the binding module remains fixed and is not the main bottleneck anymore.

## A.3 SCENE GRAPH PARSING DISCUSSION

**Comparison of different parsing methods** Although the parsing method is not the core of our contribution we provide here a couple of qualitative and quantitative comparisons to motivate the choice of using an LLM to perform the parsing of the captions despite the pre-processing computational overhead it entails. We identify 3 families of parsing method that operate on text-only input and provide insights on their respective :

- **Automatic parsing methods** : method based on hand-crafted rules about the semantics in order to extract tags and more complex dependency graphs. TagAlign also compares to nltk and justifies the choice of going to an llm-based method. We consider a representative of those automatic parsing methods based on spacy (Honnibal and Montani, 2017).

- **Finetuned factual scene graph parser** trained in a supervised way to extract scene graph. We consider a representative of them, a state-of-the-art factual scene graph parser based on T5 model (Li et al., 2023b) trained to extract fine-grained scene graph information about the objects and relations in an input caption.

- **LLM-based**, here we choose llama3-8b as a representative and leave the extensive analysisof the bias/cues of different llm families of model for future work.

We identified failures modes of automatic parsing and finetuned that are relevant to compositional understanding of clip-like models and justify the use of an llm-based parsing method and summarize them in Table 10. We show on one hand that automatic parsing methods are prone to oversimplification, missing relations and mistaking an attribute modifiers with an object. On the other hand supervised scene graph parser seems to be prone to relation classification error and important atteibute binding error when the different objects mentioned in a caption share the same label tag.

| Caption | Spacy | T5 | LLM |
|---------|-------|-----|-----|
| A brown cat is lying on a computer | Objects: a brown cat, a computer
Relations: {on, 0, 1}  (Oversimplification error) | Objects: brown cat, computer
Relations: {lay on, 0, 1}  (Relation classification error) | Objects: brown cat, computer
Relations: {lying on, 0, 1} |
| A man is on the left of the dog | Objects: a man, the left, a dog  (Wrong POS)
Relations: {of, 1, 2}  (Missing relation) | Objects: man, dog
Relations: {at the left of, 0, 1} | Objects: man, dog
Relations: {on the left of, 0, 1} |
| A woman in blue and a woman in red | Objects: a woman, red, a woman,  (Wrong POS)
Relations: {, 0, 1}, {in, 0, 2}, in, 2, 3} | Objects: blue red clothes, woman  (Wrong attribute binding)
Relations: {wear, 0, 1} | Objects: woman in red, woman in blue
Relations: {} |

Table 10: Comparison of parsing errors made by different parsers.

We additionally train OC-CLIP on COCO captions parsed by those 3 different parsing models and compare the downstream compositional understanding performance in Figure 5. Coherent with the qualitative analysis the choice of the parsing family mostly impact relational understanding. We observe for the SugarCrepe swap object (replace rel resp.) a decrease of 9.3% (resp. 14.1%) for spacy and 3.4% (resp. 6.3%) for a supervised T5 model as compared to OC-CLIP on scene graphs extracted by llama3-8b. Close to our work, TagAlign(Liu et al., 2024b) also quantitatively and qualitatively analyze the objects tags than can be extracted with an nltk-based and llm-based parser and show that training CLIP with an additional object and attribute tag classification loss with tags coming from an llm results in better downstream zero-shot semantic segemntation.

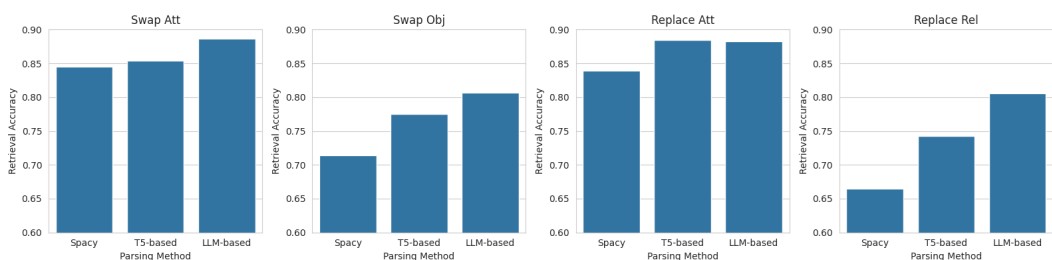

Figure 5: Downstream Compositional Understanding of OC-CLIP when trained on different parsing of COCO-Captions.

**Limitations of LLM-based parsing for OC-CLIP**   We also acknowledge that using and LLM as a parser may also have some limitations and evaluating the impact of the downstream performance of different LLMs or VLMs is an interesting question. In particular, llm-based parsing might not extract accurate scene graphs, especially when the dependency between the objects in a captions is rather complex or ambiguous. And informing the parser in prompt with visual information might be an interesting direction. However the exact instanciation of the LLM-based parser used is orthogonal to our contribution and we leave this analysis for future work.

**Scene Graph Parsing cost**   We performed the parsing by serving instances of Llama3-8b on v100 machines. Each datasets is then chunked in $N$ process that do not require any GPUs and send requests to the served LLM parsers through vllm[2] to maximize the throughput of the parallelized requests. For reference we parsed the COCO datasets ($\sim$ 500k captions) parallelizing 10 instances of the parser, and with 128 chunks in 3.5 hours and Visual-Genome ($\sim$ 200k captions) with 8 instances, 64 chunks in 1.7hours. The parsing time can further be optimized by serving more instances, using more performant GPUs (A100, H100 etc..), serving each instance in parallel in more GPUs to maximized the number of requests that can be processed per second.

## A.4   IMPORTANCE OF SYNTHETIC EXPERIMENTS

The rise of data-centric hard negative methods were motivated by the bag-of-words behaviour (Yuksekgonul et al., 2023b) of CLIP noticed in "simple swap-attribute" retrieval tasks. Hard-negative methods propose to mitigate this behaviour by finetuning CLIP-like models on data points with minimal changes but semantically different meanings. However we experimentally observed that all the methods fail to increase performance specifically in swap attribute kind of splits. In order to further isolate the root cause, we propose a series of synthetic experiments that compare covering more hard-negative data points with OC-CLIP on varying proportion of training samples and hard-negative samples. By restricting the environment to a closed-set vocabulary of backgrounds, attributes, and object classes, we can enumerate all possible hard-negatives, allowing us to systematically evaluate the effectiveness of different approaches. Our results show that simply *adding more hard-negatives plateaus and is not sample-efficient, as the swap attribute binding performance always underperforms OC-CLIP trained on less data without any hard-negatives* in a simple object-attribute binding task 2. However, when combined with OC-CLIP inductive bias, hard-negatives complementarily improve downstream performance. This suggests that our model, OC-CLIP, is a more sample-efficient approach to addressing the bag-of-words behavior of CLIP models. We hypothesize that the root cause of this issue thus lies in the representation format used in CLIP's original formulation, which relies on a single vector to capture complex semantic relationships. Our proposed method introduces inductive biases that allow the model to learn more structured representations, avoiding superposition of features (Greff et al., 2020b) and effectively mitigating the bag-of-words behavior. Through these synthetic experiments, we demonstrate the effectiveness of our approach and provide insights into the sample-efficiency limitations of existing data-centric methods.

---

[2]https://github.com/vllm-project/vllm

## A.5 PUG DATASET

In this section we describe in details the content of the controlled 3D environment based on PUG (Bordes et al., 2023). We operate in a 3D envrionment with pairs or single textured animals in different backgrounds. The factors of variation are :

- **5 Backgrounds** : desert, arena, ocean floor, city, circus
- **20 Animals** : goldfish, caribou, elephant, camel, penguin, zebra, bear, crocodile, armadillo, cat, gecko, crow, gianttortoise, rhinoceros, dolphin, lion, orca, pig, rabbit, squirrel
- **4 textures** : red, white, asphalt, grass
- **2 spatial constraints** for pairs : left/right, above/under

The different splits We then construct splits that aim at evaluating separately attribute binding and spatial relationships understanding. In all the different splits, we include images with single animals in all the possible *background-texture-animal* conjunctions.

**Attribute Binding Splits** The attribute binding training and testing splits are constructed as follows : (1) - We list all the possible pairs of animals,(2) - We randomly and i.i.d. select a percentage % $N_{\text{train}}$ of pairs to include in the train split, (3) - For each training pair we select a pair of assigned attribute (for example if cat and caribou are in the train split we will assign red to cat and white to caribou and will remove all the other attribute-animal conjunction from the training. This is done such that we can control for the *replace attribute* hard negative presence. (4) - For each pair in the training set we separate the corresponding hard negative examples with the same bag of words but swapped attributes (referred to as *Seen Pairs* in Figure 6) and the same pair but a different bag of words ( referred to as *Different Bag-of-words* in 6) , (5) - finally we also isolate unseen pairs of animals. We also include the accuracy on the training pairs that do not have their corresponding hard negatives in the test set).

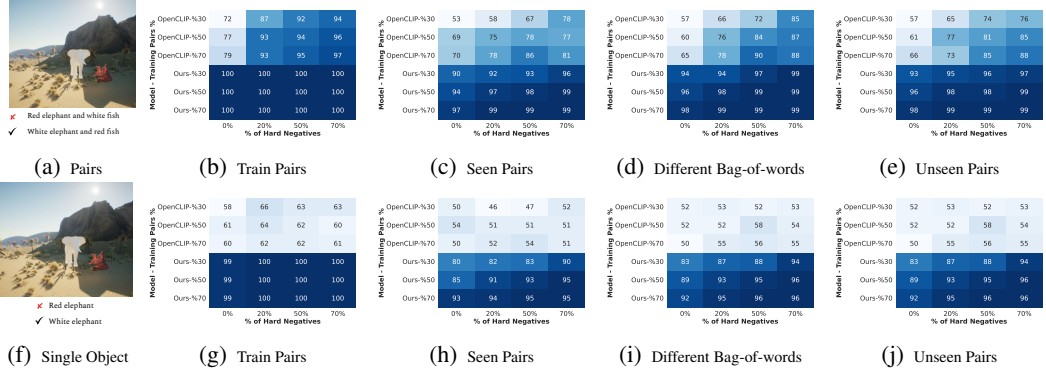

(a) Pairs     (b) Train Pairs     (c) Seen Pairs     (d) Different Bag-of-words     (e) Unseen Pairs

(f) Single Object     (g) Train Pairs     (h) Seen Pairs     (i) Different Bag-of-words     (j) Unseen Pairs

Figure 6: **Attribute Binding on PUG - Additional Results** Performance of the finetuned OpenCLIP and OC-CLIP models on a binary classification task between a caption and its corresponding hard-negative. We do that for captions that mention Pairs of animals (**top row**) like the example in Figure (a) and for captions that mention a single animal (**bottom row**) like the example in Figure (b).To assess the models' performance, we compute the accuracy across two dimensions. The first one is the percentage of animal pairs (y-axis) seen during training (animals like elephants and fish could be seen either alone or with other animals but never together). The second dimension (x-axis) is the number of hard-negatives used in the training data. For instance, whether we have the combination "red elephant" and "white fish" in the training data while we only have "white elephant" and "red fish" in the test data.

**Spatial Relation understanding Splits** For these splits we do not assign specific pairs of attributes to train/test split but rather consider pairs of animals and their order with respect to the spatial relationship tested and systematically include all the possible attributes assignment to those pairs. We then construct the different splits by restricting the number of pairs and their spatial configuration.

**Hard Negative Samples** For both tasks the hard negative samples we consider are align with the test tasks taxonomy. For attribute binding we always test the model's ability to distinguish between eg. *a red cat and a white caribou* and *a white cat and a red caribou*. Hence we consider as a hard negative sample any image that corresponds to the swapped attribute version of a training pairs. To augment the dataset with hard negative, we sample i.i.d. a percentage % $N_{\text{hard}}$ of the training pairs and include in their corresponding hard negatives in the train set. Similarly for the spatial relationship understanding task, we test the model's ability to distinguish between eg. *a red cat to the left of a white caribou* and *a white caribou to the left of a red cat*. Hence we consider as a hard negative sample any image that corresponds to the swapped order with respect to the relationship tested of the animal pairs seen during training.

### A.5.1 SPATIAL RELATION UNDERSTANDING

In this section, we aim to evaluate the spatial relationship understanding capabilities of the models. To do so, we conduct controlled experiments using data splits where not all pairs of animals are seen during training. The relations considered in these experiments are "left/right" and "above/below". Hence, the task is to choose between the original caption of the form "X left of Y" and the caption with the swapped order "Y left of X". We consider the following generalization axes:

- **Unseen object order:** This axis tests the generalization when swapping the order of objects in a relationship. For example, "elephant to the left of fish" may be used for training, while "elephant to the right of fish" is used for evaluation

- **Unseen object pairs:** This axis test for unseen pairs of animals in seen relationships.

We follow the experimental setup of section **??**, and finetune OpenCLIP and OC-CLIP while considering the effect of adding different % of hard negative images and/or different % of object pairs to the training data.

We test both models on image-text retrieval tasks and report the results in Figure 7. Figure 7(b) shows the results for the unseen object order generalization, whereas Figure 7(c) presents the results for the unseen object pairs. As shown in Figure 7(b), OC-CLIP outperforms OpenCLIP in all data regimes considered, with improvements between 6% and 18%. Similarly, as shown in Figure 7(c), OC-CLIP improves upon OpenCLIP in all data regimes, yielding absolute improvements between 5% and 20%.

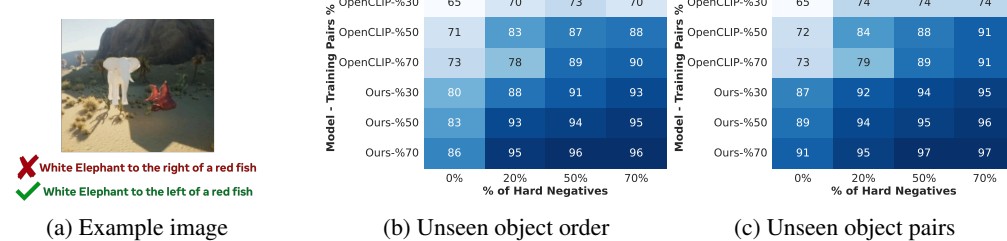

|  (a) Example image  |  (b) Unseen object order  |  (c) Unseen object pairs  |

Figure 7: **Spatial Relationship Understanding.** We finetune both OpenCLIP and OC-CLIP on splits containing different % of animals pairs (y-axis) and different % of hard-negative image in the training split (x- axis). We test the models on images with either unseen order (b) or unseen pairs (c) during training. The testing is done against the swapped order of the ground truth caption as shown in the visual example (a).

### A.6 PARSING

For the parsing of the training and testing data we used a llama-3-70b Instruct model with the following prompt :

---

**Parsing Prompt**

Given a caption, your task is to parse it into its constituent noun phrases and relationships. The noun phrases should represent independent visual objects mentioned in the caption without semantic over-simplification. For each caption, output the parsed noun phrases (e.g., entities) and relationships in JSON format, placing the dictionary between `[ANS]` and `[/ANS]` brackets. In the relationships, use indices to specify the subject and object of the relationship mentioned in the caption. The indices of the subject and object should be integers. Here are a few examples:

```
Caption: A large brown box with a green toy in it
Output:
[ANS]
{
  "entities": [
    "large brown box",
    "green toy"
  ],
  "relationships": [
    {
      "relationship": "in",
      "subject": 1,
      "object": 0
    }
  ]
}
[/ANS]

[...]  More examples
```

PAY ATTENTION to the following:
- Relationships MUST relate two different entities in the caption and NOT be unary. For example, in the caption 'red suitcases stacked upon each other', 'stacked upon each other' is not considered a relationship.
- Do not forget any relationships.
- Relationships MUST be directed. 'and' is not a relationship.
- Pay attention to spatial relationships like 'behind', 'left of', 'with', 'below', 'next to', etc. 'and' is not a relationship.
- Check the right dependencies when the relationships are not direct. In the caption template a X with a Y in it, it refers to X.
- Pay attention to co-references.

Now, parse the following caption into its constituting entities and relationships. You MUST place the answer between `[ANS]` and `[/ANS]` delimiters.
Caption:

---

## A.7 DATASETS

**Training Data** For the compositional experiments we train both OpenCLIP and OC-CLIP on a aggregated data form COCO-Captions (COCO) (Lin et al., 2014), Visual Genome (VG) (Krishna et al., 2017) and GQA (Hudson and Manning, 2019). All these datasets cover the same 110k images from COCO but focus on different kind of annotations. COCO provide global scene annotation, Visual Genome emphasizes specific region descriptions and general relationships and GQA annotates both objects and spatial relationships. Both Visual Genome and GQA have annotated scene graph that we do not need to parse to train OC-CLIP. For OpenCLIP, we sample 2 region annotations from VG to from a caption following this template *A photo of a {Region 1} and a {Region 2}*. Similarly to get the captions from GQA, if there is a relationship we follow Kamath et al. (2023) and give the model a caption following this template *A photo of {Subject} {Rel} {Object}*. If only objects are mentionned we sample up to 3 objects and give the model a caption following this template *A photo of {Obj1},{Obj2},{ Obj3}* .

## A.8 TRAINING DETAILS AND HYPERPARAMETERS

In table 11 we detail the hyperparameters of the OC-CLIP architecture.

**Optimization Details** In order to train OC-CLIP we followed prior work and use Adam Optimizer with $\beta_1$ and $\beta_2$ set to 0.9 and 0.95 and a weight decay of 0.2. We used different learning rate for the pretrained backbones and for our modules that we train from scratch : learning rate of $2e^{-4}$ for the binding and the scoring modules, learning rate of $2e^{-5}$ for the text Transformer backbone, and a smaller rate of $1e^{-6}$ for the ViT backbone. We also used a warmup schedule for both of the text (1k steps) and the vision (5k steps) backbones followed by a cosine decay. We train the model for a total of 100 epochs.

| Hyperparameter/Parameter Init | Architecture | Value |
|---|---|---|
| **Binding Module** | | |
| – Image Patches Processing | MLP(per patch) | in $\times$ 256 |
| – Self-Attention #Layers/#Heads | | 2/4 |
| – Self-Attention MLP ratio/act | | 2/nn.GELU |
| – Keys $K$, Values $V$ | Linear | 256, 256 |
| – Normalization Keys/Values | LayerNorm | 256 |
| **Grouping Module** | | |
| – Cross-Attention #Heads | | 1 |
| – Queries | Linear | 256 |
| – Normalization Queries | LayerNorm | 256 |
| – Num Default Tokens $Q_{\text{default}}$ | nn.Param($N_d$,256) | 4 |
| **Scoring Functions** | | |
| – Object Scoring Function | cosine sim | |
| – Relation Scoring subject $f_s$ | MLP(128 + 256, 128) | 2 layers |
| – Relation Scoring object $f_o$ | MLP(128 + 256, 128) | 2 layers |
| – Coef ent init (learned parameter) | | 1.5 |
| – Coef rel init (learned parameter) | | 0.5 |

Table 11: Table of hyperparameters for OC-CLIP architecture

## A.9 ATTENTION MAPS

See Figure 8

## A.10 BINDING MODULE CODE

See Figure 9

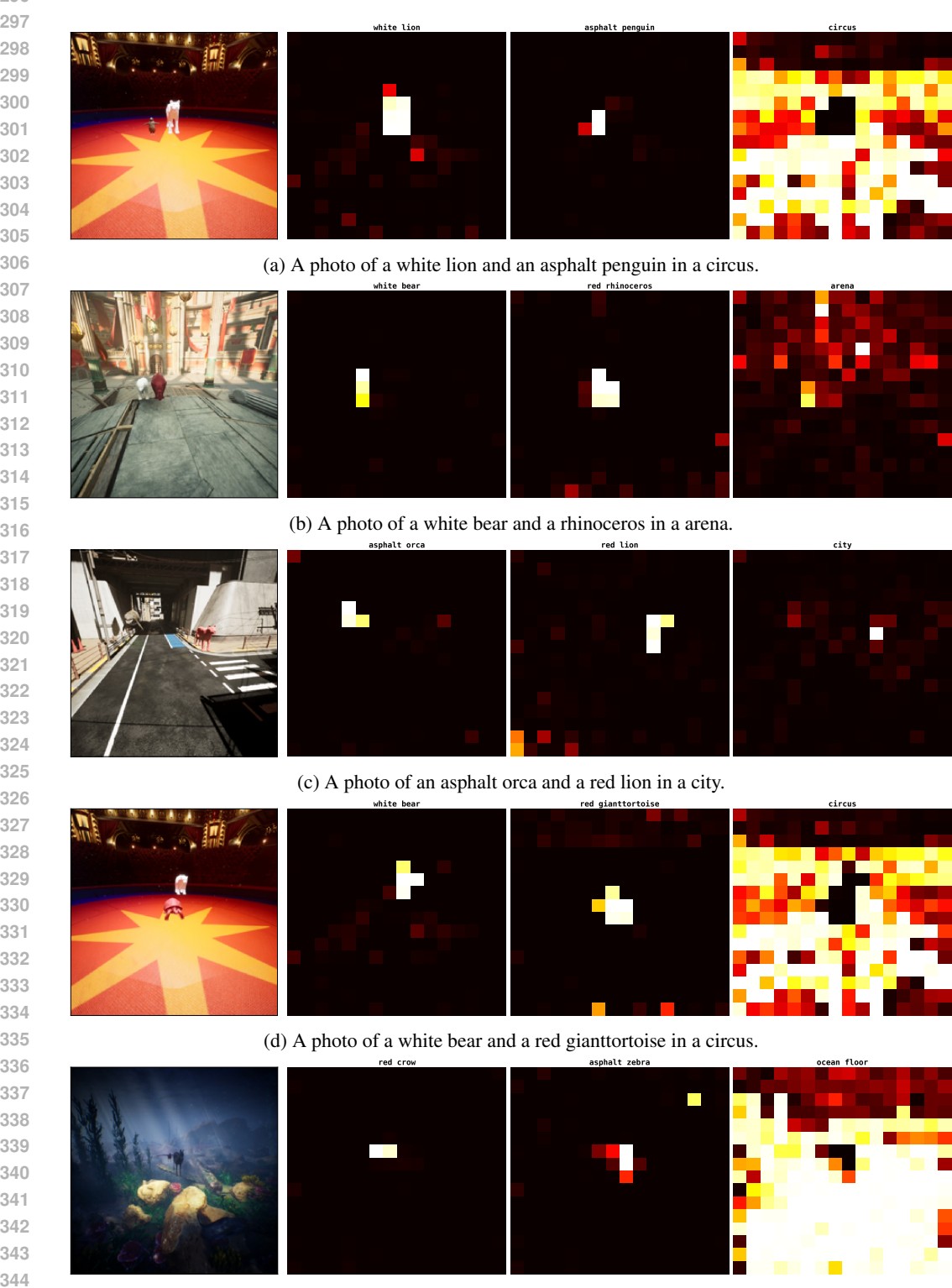

(a) A photo of a white lion and an asphalt penguin in a circus.

(b) A photo of a white bear and a rhinoceros in a arena.

(c) A photo of an asphalt orca and a red lion in a city.

(d) A photo of a white bear and a red gianttortoise in a circus.

(e) A photo of a red crow and an asphalt zebra in a ocean floor.

Figure 8: OC-CLIP Binding Attention Maps on PUG. We plot the attention maps of each query object in the caption (specified at the top of each attention map) and notice that natural objects emerge.

```python
class AssignAttention(nn.Module) :
    def __init__(
        self, dim, qkv_bias=False, qk_scale=None):
        super().__init__()
        self.scale = qk_scale or dim**-0.5
        self.q_proj = nn.Sequential(nn.Linear(dim, dim, bias=qkv_bias))
        self.k_proj = nn.Sequential(nn.Linear(dim, dim, bias=qkv_bias))
        self.v_proj = nn.Sequential(nn.Linear(dim, dim, bias=qkv_bias))

    def forward(self, query, key=None, value=None):
        #before cross attention projections
        q = self.q_proj(query)
        k = self.k_proj(key)
        v = self.v_proj(value)
        #scaled dot product
        attn = (q @ k.transpose(-2, -1)) * self.scale
        #softmax across query dim
        attn_dim = -2
        attn = F.softmax(attn, dim=attn_dim) + 1e-8
        #attn normoalization
        attn = attn / (attn.sum(dim=-1, keepdim=True))
        output = torch.einsum("bqk,bkd->bqd", attn, v)
        return output
class BindingModule(nn.Module) :
    def __init__(self,in_vis_dim, dim, num_patches, num_default_tokens) :
        super().__init__()
        self.im_proj = nn.Sequential(nn.Linear(in_vis_dim, dim),
        nn.GELU(), nn.Linear(dim, dim))
        self.pos_embeddings = nn.Parameter(torch.randn(num_patches, dim))
        self.img_processor = nn.Sequential( ResidualAttnBlock(dim, 4),
        ResidualAttnBlock(dim, 4))
        self.default_tokens = nn.Parameter(
        torch.randn(1, num_default_tokens, dim))
        self.to_kq_groups = nn.Sequential(nn.Linear(dim, 2 * dim))
        self.dim = dim
        self.num_default_tokens = num_default_tokens
        self.k_norm = nn.LayerNorm(dim)
        self.v_norm = nn.LayerNorm(dim)
        self.assign_slots = AssignAttention(dim)

    def encode_patches(self, patches) :
        patches = self.im_proj(patches)
        patches = patches + self.pos_embeddings
        patches = self.img_processor(patches)
        K_img, V_img = torch.split(
        self.to_kq_groups(patches),self.dim,dim=-1)
        K_img, V_img = self.k_norm(K_img), self.v_norm(V_img)
        return K_img, V_img

    def group(self, query_tokens, K_img, V_img) :
        #adding default tokens
        query_tokens= torch.cat([query_tokens,default_tokens ], 1)
        out = self.assign_slots(query_tokens, K_img, V_img)
        #remove default tokens
        out = out[:, :-self.num_default_tokens]
        return out
```

Figure 9: Code for the Binding Module

