# OpenReview forum: "OC-CLIP : Object-centric binding in Contrastive Language-Image Pretraining"
_ICLR.cc/2025/Conference — Submitted to ICLR 2025_

### Official Review · Reviewer_HgdJ · 2024-10-16

**Soundness:** 3
**Presentation:** 2
**Contribution:** 2
**Rating:** 5
**Confidence:** 5

**Summary:**

Most existing vision and language models (VLMs) face challenges when dealing with compositional reasoning in images and texts. To enhance the compositional reasoning capabilities of VLMs, previous research primarily refined these models through the construction of specialized datasets, such as introducing hard negative examples via textual modifications. This paper presents an innovative approach aimed at overcoming the limitations of existing VLMs when handling complex compositional scenes involving multiple objects and their spatial relationships. The work introduces a novel framework focused on integrating sufficient inductive biases into pre-trained CLIP-like models, providing additional guidance through the use of scene graphs rather than relying on traditional hard negative augmentation strategies. Experimental results indicate that such a design optimizes the performance of CLIP models across multiple benchmark datasets.

**Strengths:**

The idea presented in this paper of using scene graphs to help VLMs improve their compositional reasoning capabilities is novel and interesting, and it appears to be an effective approach.

This paper proposes a new structured similarity score based on contrastive loss, consisting of an object scoring function and a relationship scoring function, which seems capable of effectively evaluating the relationships between nodes in the scene graph.

The organization of this paper is clear, the problem definition is explicit.

The proposed method has achieved effective improvements on certain datasets.

**Weaknesses:**

This paper lacks convincing evidence in several key areas, specifically:

The experimental comparisons are clearly insufficient. Mainstream datasets in this field, such as the CoCo and Flickr datasets from the ARO dataset, the VL-Checklist dataset, and the sDCI dataset, have not been included in the comparative experiments. Additionally, mainstream methods such as DAC, sDCI, and SVLC have not been compared, making the experimental results significantly unconvincing.

VL-Checklist: "Vl-checklist: Evaluating pre-trained vision-language models with objects, attributes and relations." arXiv preprint arXiv:2207.00221 (2022).
sDCI: "A picture is worth more than 77 text tokens: Evaluating clip-style models on dense captions." Proceedings of the IEEE/CVF Conference on Computer Vision and Pattern Recognition. 2024.
DAC: "Dense and aligned captions (dac) promote compositional reasoning in vl models." Advances in Neural Information Processing Systems 36 (2024).
SVLC: "Teaching structured vision & language concepts to vision & language models." Proceedings of the IEEE/CVF Conference on Computer Vision and Pattern Recognition. 2023.

Ablation studies are missing. There is no analysis of the ablation of the loss function or the selection of object and relationship extraction models.

The accuracy of the scene graph heavily relies on the relationship extraction model. However, models like Llama3 also face issues with misalignment in compositional reasoning. This paper does not provide any method to ensure the accuracy of the scene graph.

There is no analysis of whether CLIP retains its original image understanding capabilities after fine-tuning with the scene graph.

The proposed loss function in the paper includes multiple hyperparameters, but there is no experimental or theoretical analysis provided.

**Questions:**

Experimental Comparisons:
1. Why not include comparisons with mainstream datasets like CoCo, Flickr (ARO dataset), VL-Checklist, and sDCI?

2. Why are mainstream methods such as DAC, sDCI, and SVLC not included in your experiments?

Ablation Studies:
3. Provide an analysis of the ablation of the loss function.

4. Include an ablation study on the selection of object and relationship extraction models.

Scene Graph Accuracy:
5. How do you address the issue of misalignment in compositional reasoning, as seen in models like Llama3?

6. What methods do you propose to ensure the accuracy of the scene graph?

CLIP Fine-Tuning:
7. Analyze whether CLIP retains its original image understanding capabilities after fine-tuning with the scene graph.

Loss Function Hyperparameters:
8. Provide experimental or theoretical analysis to justify their inclusion and effectiveness.

---

> ### Author Response · Authors · 2024-11-22
> **Rebuttal Answer - Part 1**
>
> We thank the reviewer for their time and their extensive feedback. We believe we improved our work considerably  and address the concerns in the following.
>
> We made the following additions that we further detail below :
>  * Additional Benchmarks (VL-Checklist + full ARO)
>  * Additional Baselines (SVLC,  DAC, DCI)
>  * Extensive modules + loss ablation study
>  * Ablation of different parser families
>  * Training from scratch of CLIP and OC-CLIP on CC12M and zero-shot classification evaluation.
>
>
> **Additional Experimental Comparisons**
>
> Our goal is to evaluate CLIP-like models' compositional understanding in **plausible and grammatically correct cases**. The SugarCrepe benchmark has identified exploitable textual biases in previous mainstream procedurally-generated hard negatives benchmarks like the COCO and Flickr set of ARO and VL-checklist. Specifically, \[1\] shows that procedurally generated hard negatives are either highly grammatically incorrect and can be identified by a blind model or by a good language model. SugarCrepe proposes to follow the same fine-grained taxonomy on attributes, objects, relationships  as VL-checklist but ensures that the hard-negative are not distinguishable by a blind model. However, as requested by the reviewer we **added the results on the full ARO suite and VL-Checklist in Table 5**. We also **added DAC, DCI, SVLC as suggested as baselines in sugarcrepe and ARO/VL-checklist**. As expected these methods still fall behind in splits that require specific object-attribute binding :
>
> |  | OC-CLIP | DAC-LLM | DAC-SAM | DCI | CLIP-SVLC |
> | :---- | :---- | :---- | :---- | :---- | :---- |
> | Swap-Att | 88.5 | 74.1 | 75.3 | \- | \- |
> | ARO-A | 82.0 | 73.9 | 70.5 | 67.6 | 73.0 |
>
>
> **Ablation Study**
>
> We appreciate the reviewer's feedback and have addressed the concerns regarding ablation studies. In response to points 3 and 4, **we have added an extensive ablation section (Appendix A.1)** that thoroughly examines the specific object-centric design choices related to OC-CLIP.
>
> In particular, our analysis includes an ablation study of the loss function and we investigate the impact of the following key components: Local loss, competitive attention, default token, relation scoring module modularity.
>
> These ablations provide valuable insights into the importance of each component in the overall performance of our model. Most notably removing the local loss (as shown in Table 6\) effectively impacts downstream relational understanding on SugarCrepe with a swap obj accuracy decreasing from 80.7 to 73.1 and a replace rel accuracy decreasing from 80.6 to 74.7.
>
> We believe that this additional analysis strengthens our paper and provides a more comprehensive understanding of the OC-CLIP architecture. We hope that this revised version addresses the reviewer's concerns.
>
> **Scene Graph Accuracy**
>
> We want to stress the fact that the specific LLM instantiation is not a claimed contribution but rather a tool used in the proposed method.  In particular, we consider an LLM-based parser as the best automated option currently available for obtaining scene graphs.
>
> To support this claim, we **have added a quantitative and qualitative ablation study (Appendix A.3)** that compares three different parser families:
>
> * LLM-based parsers (llama3)
> * Automatic parsing based on Spacy
> * Supervised scene graph parsers (T5-based)
>
> This qualitative and quantitative ablation study provides valuable insights into the strengths and weaknesses of each parser family, illustrating some important failure modes (**see added Table 9 in the Appendix**)  of automatic and supervised scene graph parsers that impacts downstream compositional performance when OC-CLIP is trained on those scene graphs (**see added Figure 6 in the Appendix**). We notice the strength of the parser impacts the relational understanding performance with a swap obj (llm) : 80.7 (supervised) :77.5, (automatic) : 71.4 and replace rel (llm) 80.6, (supervised) 74.3, automatic (66.5) which is consistent with the qualitative failure modes identified in Table 9\.
>
> [1] SugarCrepe : Fixing Hackable Benchmarks for Vision-Language Compositionality

---

> ### Author Response · Authors · 2024-11-22
> **Rebuttal Answer - Part 2**
>
> **Analysis of OC-CLIP’s general understanding capabilities**
>
> There is a misunderstanding on the fine-tuning point highlighted by the reviewer. There is **no fine-tuning of the binding module involved** in our method. We are training the binding module **from scratch** on top of a pre-trained CLIP backbone and comparing it to methods that do finetuning and **do not add any additional parameters**. Regarding the performances of the binding module itself, it is strictly bound by the vocabulary seen during its training (eg. COCO/VG/GQA) and cannot be expected to generalize further. The only way to assess correctly the differences between CLIP and OC-CLIP in both general and compositional image understanding abilities is to train CLIP and OC-CLIP from scratch on the same training data. To that end, **we added CC12M experiments where both backbones are trained from scratch**. Notably here the scene graphs are obtained from a noisy and non-human curated dataset. We added those results in Table 4 and 7 and  noticed improvements in general understanding with **\+9.2% on Imagenet zero-shot classification** performance while maintaining a significant gap in **zero-shot sugarcrepe swap attribute (+15.9%) and swap obj (+14.3%) splits**. We added the extensive zero-shot classification results on the **ELEVATER suite in Table 7** [1].
>
> **Analysis of Loss Function Components**
>
> The proposed loss function includes multiple components, including the local graph contrastive loss, which is designed to prevent the relational module from collapsing and ignoring visual slots. We provide a **detailed motivation for adding this component in Appendix A.1. (paragraph Local Graph contrastive loss)**
>
> Regarding the weighting components, specifically α and β, we clarify that these are learned parameters, not hyperparameters. This means that they are automatically adjusted during training to optimize the model's performance. We clarified the choice of initial value in Appendix A.1 (paragraph Similarity Score).
>
> To further demonstrate the importance of the local loss component, we provide an **additional experiment (Table 6\) where we remove the local loss** and evaluate the impact on relational understanding performance. The results show that removing the local loss drastically impacts relational understanding performance **(-7.5% and \-5.9% in the swap obj and replace rel sugarcrepe splits)** highlighting its effectiveness in the overall model.
>
> We hope we have addressed the reviewer concerns with those additional experiments and are happy to answer to any further questions the reviewer might have.
>
> [1] ELEVATER : ELEVATER : https://computer-vision-in-the-wild.github.io/ELEVATER/

---

> > ### Comment · Reviewer_HgdJ · 2024-11-26
> > **from 3 to 5**
> >
> > Thank you for your feedback. Many of my concerns have been addressed, so I am prepared to increase my score to 5. I appreciate this work, but I still believe it has not addressed the most critical issue—how to improve the quality of compositional training data. Additionally, I remain skeptical about the method of representing relationships using patches. I think there is still a lot of room for improvement in this work, and the current experiments do not sufficiently demonstrate its effectiveness.

---

> > > ### Author Response · Authors · 2024-11-26
> > > **relationships clarification**
> > >
> > > We thank the reviewer for their feedback. Could the reviewer give us some pointers about the non-effectiveness of our relationship module ?
> > >
> > > To be a bit more precise, the relationships component are obtained from object-centric visual slots (so 1 vector per object mentioned in the caption, which is some leaned grouping of patches). A relationship is described by a predicate and is applied to two entities (the subject and object of this relationship).  In the following caption "A dog to the left of a cat" It seems rather intuitive that the relationship contribution to the total similarity score should depend on the visual representation of the dog, the visual representation of the cat and the nature of the predicate "to the left of".
> > > We additionally gave strong quantitative evidence that this way of modeling relationships work, specifically on spatial relationship understanding benchmark where none of the other methods are efficient. See Table 2 that we paste here :
> > > | Model | OC-CLIP | CLIP | NegCLIP | XVLM | BLIP |
> > > | :---- | :---- | :---- | :---- | :---- | :---- |
> > > | COCO-spatial | 93.5 | 45.6 | 46.4 | 73.6 | 56.4 |
> > > | GQA-spatial | 95.6 | 49.1 | 46.7 | 67 | 52.6 |

---

> > > > ### Author Response · Authors · 2024-11-28
> > > >
> > > > We thank the reviewer again. We would love to further discuss their concerns regarding the relationship module.

---

### Official Review · Reviewer_uY5M · 2024-10-30

**Soundness:** 2
**Presentation:** 3
**Contribution:** 2
**Rating:** 5
**Confidence:** 4

**Summary:**

The paper proposes augmenting CLIP contrastive finetuning with a "binding module" supervised by objects and relations data extracted from high-quality human-annotated data via an LLM. The proposed architecture of the binding module explicitly matches objects and relations to learned attention slots applied to visual encoder output. Noise register tokens are also used in the matching. Experiments were conducted on either synthetic data (train and test generated by the authors) or training on high-quality real VG / COCO / GQA annotated data. Testing of the real data-trained models is primarily done on ARO, SugarCrepe, and GQA (which are derived from the same datasets used for training). Additionally, WinoGround performance improvements are reported. Seems some important baselines are missing.

**Strengths:**

* compositional reasoning is still an interesting problem, yet as the field has mostly shifted to decoder LMMs it is unclear how long the encoder-only models can still be compared just to themselves. Decoder LMMs have much higher compositional reasoning performance.
* designing and training explicit alignment modules for objects and relations is interesting
* performance improvements reported

**Weaknesses:**

1. synthetic data experiments seem limited and less convincing.
2. in the real data experiments, the model is finetuned on high-quality human annotated data from the same datasets out of which the evaluation benchmarks (ARO, SugarCrepe, COCO) are constructed. It seems that the method performs supervised training as opposed to hard-negative based techniques mentioned which train on unconstrained internet data (LAION, CC, etc)
3. seems that important baselines are missing as well as additional compositional reasoning benchmarks to compare to those baselines.
e.g. https://paperswithcode.com/sota/visual-reasoning-on-winoground - for many higher reported results on WinoGround; more hard negative mining methods (DAC = https://arxiv.org/pdf/2305.19595 being one example); etc
4. no zero-shot evaluation of the fine-tuned model is done - the model is fine-tuned for 100 epochs on supervised data explicitly focusing objects and relations there - one might expect significant degradation on zero-shot benchmarks like ELEVATER used in DAC above. It is important to show the proposed technique does not solve some compositional reasoning issues on expense of the general applicability of the model (zero-shot performance)
5. Ablation of the proposed components and losses is completely missing, but should be a significant part of the paper informing the reader on the relative contribution of components, insights related to them, and so on.
6. More minor:
a. the "rel" loss is very similar to the hard negative losses employed by others, a bit counter the (a bit unjustified) reasoning of the authors that hard negative based approaches are detrimental (if anything they require much less supervision than the proposed approach) - also rel loss is not ablated (well nothing is ablated except the synthetic dataset)
b. reliance on LLM parsing is not ablated, what if it introduces noise how to deal with it? Parsing real captions (eg if one would try to apply this on unconstrained data like LAION or CC) seems a much harder task compared to negative augmentation used in other works (contrary to the reasoning of the authors in the intro)
c. in eq. 3 - what prevents the collapse of f_s and f_o to just extracting the "r" part from the concat? not ablated / analyzed...

**Questions:**

please see weaknesses, lots of questions there

---

> ### Author Response · Authors · 2024-11-22
> **Rebuttal Answer - Part 1**
>
> We thank the reviewer for all those precise comments/questions and address them in the following.
>
> We made the following additions that we further detail below :
>  * Clarification of the goal of the synthetic data
>  * Training from scratch of both CLIP and OC-CLIP on cc12m and evaluation on zero-shot classification from ELEVATER
>  * Additional baselines (svlc, dac, dci)
>  * Extensive ablation study on the model + loss design choice
>  * Ablation on different parser families
>
> **Synthetic Data Experiments**
>
> We thank the reviewer for this suggestion. It’s true that the synthetic experiments have taken too much space in the first paper version in contrast to real world experiments. To have a better balance between synthetic and real experiments, **we moved part of the synthetic experiments now to the Appendix A.5 and added new experiments on the CC12M dataset instead (see later comments)**.
>
> The primary goal of the synthetic data experiments was to **isolate the root cause of the bag-of-words behavior in CLIP models** and evaluate the effectiveness of different approaches in addressing this issue. Those experiments allowed us to make the following observations about the use of hard-negatives:
>  * (1) simply adding more hard-negatives plateaus in terms of model accuracy, and
>  * (2) it is not sample-efficient, even in an easy synthetic environment.
> Notably, *the attribute-binding results on synthetic data (Table 1\)* are consistent with the gap in swap attribute performance on SugarCrepe of  OC-CLIP and both small-scale (NegCLIP, CE-CLIP ..) and large-scale (DAC, DCI, CLIP-SVLC that we added as requested) hard-negative methods. Our results show that all methods plateau and do not see any improvement, with OC-CLIP achieving over 88%.
>
> **General Understanding Capabilities**
>
> There is a misunderstanding on the fine-tuning point highlighted by the reviewer. There is **no fine-tuning of the binding module involved** in our method. We are training the binding module **from scratch** on top of a pre-trained CLIP backbone and comparing it to methods that do finetuning and **do not add any additional parameters**. Regarding the performances of the binding module itself, it is strictly bound by the vocabulary seen during its training (eg. COCO/VG/GQA) and cannot be expected to generalize beyond the seen vocabulary. The only way to assess correctly the differences between CLIP and OC-CLIP in both general and compositional image understanding abilities is to train CLIP and OC-CLIP from scratch on the same training data. To that end, we added CC12M experiments where both backbones are trained from scratch.  Notably here the scene graphs are obtained from a **noisy and non-human curated dataset**. We **added these results in Table 4 and 7** and noticed improvements in general understanding with **\+9.2% on Imagenet zero-shot classification** performance while maintaining a significant gap in **zero-shot sugarcrepe swap attribute (+15.9%) and swap obj (+14.3%) splits**. We added the extensive zero-shot classification results on the ELEVATER suite in Table 7\.
>
> **Additional Baselines**
>
> In our study, we focused on comparing to hard-negative-based baselines that generate hard negatives in-distribution, specifically from the COCO/VG/Flickr domain. This is why we included methods such as **NegCLIP, CE-CLIP, and CounterCurate, which also focus on generating hard negatives within these specific domains**.
> However, we acknowledge the importance of broader comparisons and **have now included additional methods such as DAC,  DCI and CLIP-SVLC  to our analysis (see Table 2 and Table 5)**. These methods are different in that they incorporate both hard negative mining on a larger scale and dense captioning/recaptioning, which is not the primary focus of our work. Nonetheless, their inclusion provides a more comprehensive comparison and allows us to better position our approach within the broader landscape of compositional reasoning benchmarks.
> Notably, we see that on the sugarcrepe/ARO splits that assess for precise object-attribute binding (swap attribute/ARO-A) all these methods still fall behind :
>
> |  | OC-CLIP | DAC-LLM | DAC-SAM | DCI | CLIP-SVLC |
> | :---- | :---- | :---- | :---- | :---- | :---- |
> | Swap-Att | 88.5 | 74.1 | 75.3 | \- | \- |
> | ARO-A | 82.0 | 73.9 | 70.5 | 67.6 | 73.0 |
>
> Which confirms the results we presented with the synthetic experiments that **the hard-negative approach, even at scale, is not sufficient to solve the bag-of-words behaviour of CLIP** and justifies the inclusion of more structured inductive biases within the architecture of the model like we do for OC-CLIP
>
> We hope that this expanded comparison addresses the reviewer's concern and provides a clearer understanding of our approach and its context within the field.

---

> > ### Author Response · Authors · 2024-11-22
> > **Rebuttal Answer - Part 2**
> >
> > **Amount of supervision**
> >
> > There is a potential misunderstanding since our method does not perform supervised training. In fact, the results we added on CC12M show that we do not need highly annotated dataset like COCO. In this experiment, OC-CLIP is still trained with a contrastive objective, and the content of the relations and objects extracted by the LLM-parser is open-ended and not restricted to a particular set. This means that our approach is more flexible and adaptable than supervised methods which come with a fixed set of categories.
> >
> > We acknowledge that our "rel" loss shares similarities with hard negative losses employed by others. However, we would like to highlight that parsing real captions, even from noisy datasets, is as challenging (if not less?) as obtaining hard negatives and re-captioning examples. Methods such as DAC, DCI, and SVLC all use an LLM that takes as input a caption and outputs either an alternative positive or negative candidate captions. This requires the underlying model to understand the semantics and scene structure in order to generate a valid candidate. When prompting the model to parse the caption we only ask the model to output its understanding of the scene structure and not come up with additional positive (more detailed, DAC/DCI) /negative(minimal change,NegCLIP,SVLC) candidates.
> >
> > **Ablations**
> >
> > We appreciate the reviewer's feedback and have addressed the concerns regarding ablation studies. In response to point 5, **we have added an extensive ablation section (Appendix A.1)** that thoroughly examines the specific object-centric design choices related to OC-CLIP.
> >
> > In particular, our analysis includes an ablation study of the loss function, where we investigate the impact of the following key components: **Local loss, competitive attention, default token, relation scoring module modularity.**
> >
> > These ablations provide valuable insights into the importance of each component in the overall performance of our model. Most notably, **removing the local loss (as shown in Table 6\) effectively harms downstream relational understanding on SugarCrepe with a swap obj accuracy decreasing from 80.7 to 73.1 and “rel” accuracy decreasing from 80.6 to 74.7**.
> >
> > We believe that this additional analysis strengthens our paper and provides a more comprehensive understanding of the OC-CLIP architecture.
> >
> > **Reliance on LLM parsing in noisy datasets**
> >
> > We want to stress the fact that the specific LLM instantiation is not part of our contribution.  Instead, we consider an LLM-based parser as the best automated option currently available for obtaining scene graphs.
> >
> > To support this claim, **we have added a quantitative and qualitative ablation study (Appendix A.3)** that compares three different parser families:
> >
> > * LLM-based parsers (llama3)
> > * Automatic parsing based on Spacy
> > * Supervised scene graph parsers (T5-based)
> >
> > This ablation study provides valuable insights into the strengths and weaknesses of each parser family, illustrating some **important failure modes (see added Table 9 in the Appendix)**  of automatic and supervised scene graph parsers that impacts downstream compositional performance **(see added Figure 6 in the Appendix)**.
> >
> > The **CC12M experiments that we added** serve as a proof of concept that our method works from scratch on **noisy datasets and has scaling potential**. We leave further scaling for future work.
> >
> > **Local rel Loss Collapse**
> >
> > In designing the structured similarity score of OC-CLIP, we formulate the relational component as a cosine similarity between the relation embedding and the sum of the subject and object embeddings. However, we acknowledge that this formulation may lead to the collapse of f_s and f_o, where they only extract the "r" part from the concatenation.
> > To prevent such collapse, we propose adding a local graph contrastive loss that shares similarities with hard-negative based learning. We enforce the model to model with a higher similarity the graph composed of the same nodes but with either swapped object and subject indices or shuffled objects and subjects indices within the local graph. This prevents the model from collapsing because the ground-truth graph is distinguishable from the perturbed graphs only if the visual representations are not ignored in the relationships components.
> > We ablate incorporating both of these perturbed graphs in Figure 4 and removing the local loss in Table 6. **Removing the local loss effectively impacts downstream relational understanding on SugarCrepe with a swap obj accuracy decreasing from 80.7 to 73.1 and rel accuracy decreasing from 80.6 to 74.7**. This demonstrates the effectiveness of the local graph contrastive loss in **preventing the collapse of f_s and f_o**.
> >
> > We hope that this analysis addresses the reviewer's concern and provides a clearer understanding of our approach.
> >
> > We are happy to answer any further questions that the reviewer may have.

---

> > > ### Comment · Reviewer_uY5M · 2024-11-25
> > > **response to rebuttal**
> > >
> > > I thank the authors for their response.
> > > To clarify:
> > > 1. The ELEVATER evaluations are intended to make sure that during the fine-tuning of strong pre-trained models like CLIP/OpenCLIP etc - they are not losing their zero-shot capabilities. However, the authors evaluated ELEVATER only for the model trained from scratch, so it kind of loosing the point... please provide the ELEVATER of OC-CLIP fine-tuned from CLIP and compared to original OpenAI CLIP, not the one trained by the authors on cc12m.
> > > 2. The OC-CLIP trained on cc12m without high-quality supervised datasets also being the basis for the evaluation benchmarks (pls see my original concern) seems to lose performance (Table 4)  in SugarCREPE Swap (Obj/Attr) compared to strong baselines in Table 1 as reported for this partition. I know Table 1 is fine-tuning and Table 4 is pre-training, yet still this somewhat illustrates that without high quality data the method seems to underperform.
> > > 3. Thanks for adding the missing ablation section, however: a. you added it in Appendix, while it should be in the main paper as it is a very important part of the paper that cannot be excluded to Supplementary that is designed for "non mandatory" read things; b. The paper is already beyond 9 pages with your updates (in blue) so this is already out of line I think and with ablation (originally missing) and now in Appendix - it puts you in unfair advantage if the paper is accepted this way.
> > >
> > > While thanking the authors for their efforts in preparing an extensive response, I still feel it would be more adequate and fair that this paper is submitted to a later venue after complementing it with improvements to 1&2 points I raised above, as well as other improvements suggested and partially executed that changed the paper much beyond its original submission, somewhat beyond I think to what can be acceptable in terms of the rebuttal / discussion process. However, I would leave the final decision on that to the AC committee. In light of all above, I currently prefer to keep my original score of 5.

---

> ### Author Response · Authors · 2024-11-25
> **reponse about finetuning**
>
> We thank the reviewer for the answer.
> **We would like to ask for some clarifications of the requests** :
>
> We  think that there is a misunderstanding in **expecting our model not losing their zero-shot capabilities**. Our model does not rely on the initial image-text alignment (from both CLS tokens of the backbones, those are **not used** anymore) but rather learns a new aligned embedding space **from scratch** and cannot be expected to retain any zero-shot capabilities. Our model is **not finetuned** but also trained from scratch for the alignment part (binding + relationship module) from which the similarity score is derived.
> Hard-negative based methods to which we compare only **finetune** an existing powerful backbone and can be expected to retain some of the original zero-shot capacities. Comparing our OC-CLIP to the originally aligned CLIP is not fair setting. **Could the reviewer explain why they think it is an informative comparison?**
>
> We put however the requested results here :
>
> | Model | Food101 | CIFAR10 | CIFAR100 | SUN397 | Cars | Aircraft | DTD | Pets | FER2013 | STL10 | EuroSAT | RESISC45 | GTSRB | KITTI Distance | Country211 | Patch Camelyon | UCF101 Frames | CLEVR Counts | Hateful Memes | Rendered SST2 | Imagenet |
> | :---- | :---- | :---- | :---- | :---- | :---- | :---- | :---- | :---- | :---- | :---- | :---- | :---- | :---- | :---- | :---- | :---- | :---- | :---- | :---- | :---- | :---- |
> | CLIP | 85.80% | 83.05% | 65.03% | 65.99% | 81.84% | 14.25% | 47.93% | 83.62% | 6.20% | 92.24% | 41.54% | 56.00% | 34.71% | 29.14% | 15.02% | 53.27% | 65.48% |12.71% | 51.80% | 54.31% | 64.57% |
> | OC-CLIP | 37.45% | 92.53% | 53.44% | 35.09% | 2.38% | 3.99% | 25.16% | 40.50% | 18.79% | 95.10% | 28.72% | 23.05% | 10.42% | 40.24% | 1.84% | 50.46% | 32.43% | 8.94% | 50.00% | 51.95% | 27.23% |

---

> ### Author Response · Authors · 2024-11-25
> **loss in performance  disagreement**
>
> We designed OC-CLIP to tackle the object-binding problem, most illustrated by the swap attribute of SugarCrepe. All the baselines we compare to start from a an already strong backbone. When OC-CLIP is trained form scratch on cc12m it rather **outperforms** all the considered baselines even those fine-tuned in domain with targeted hard-negatives. **Could the reviewer explain why they think those results are not in favor of OC-CLIP inductive biases?**
>
> See the swap attribute performance below :
> | OC-CLIP \- zero-shot (cc12m) | CLIP | NegCLIP  | DAC | CE-CLIP |
> | :---- | :---- | :---- | :---- | :---- |
> | 77.4 | 61.5 | 75.4 | 75.3 | 77.0 |
>
> NegCLIP and CE-CLIP finetune OpenCLIP on in-domain data (COCO/VG) and DAC on large-scale high-quality data (through recaptioning+ hard negative LLM-based generation), OC-CLIP here is trained from scratch on noisy data and still outperforms them.

---

> > ### Author Response · Authors · 2024-11-28
> >
> > We thank the reviewer again for their answer. We would love to hear from them regarding the two points raised above ?

---

### Official Review · Reviewer_ar8U · 2024-11-03

**Soundness:** 3
**Presentation:** 3
**Contribution:** 3
**Rating:** 6
**Confidence:** 5

**Summary:**

This paper roposes a approach that diverges from commonly used strategies relying on the design of hard-negative augmentations. This work instead focuses on integrating sufficient inductive biases into pre-trained CLIP-like models to improve their compositional understanding without using additional data annotations.

**Strengths:**

-The proposed method is simple and intuitive.

-The experimental performance on several datasets demonstrate large improvements in terms of accuracy.

-Interesting idea to generate object/relation tags by utilizing LLM to help scene graph undetstanding

**Weaknesses:**

-Using LLM for parsing and training with object + relation classification loss, are a bit too straightforward to be claimed as a contribution

-One possible way to make LLM contributions more reasonable is to compare multiple different LLM/cues and analyze the results/performance.

-Extract the computational cost of labels with LLM. LLM is known for its large scale and requires a significant amount of computing resources. I would like to know how much computation is required to process the visual genomic data used in the paper.

**Questions:**

-Can some MLLMs (e.g., internVL, llava) be regarded as text parser? Can MLLMs (use text and image) parse better relation and objects than LLM from text

-Tagalign[2] also use a LLM-based as text parser. Suggest comparing with this paper.

[2] Tagalign: Improving vision-language alignment with multi-tag classification.

---

> ### Author Response · Authors · 2024-11-22
> **Rebuttal Answer - Part 1**
>
> We thank the reviewer for their valuable feedback as they allowed us to considerably improve our work. In the following we tackle the raised points.
>
> We made the following additions that we detail below :
>  * Extensive ablations of the model design choices + losses
>  * Qualitative and quantitive ablation of different parsers families
>  * Parsing method cost detail
>  * Comparison with TagAlign
>
> **Contribution Clarification**
>
> We would like to emphasize that our contribution resides in the design of a binding module that learns to extract visual slots conditioned on open-ended object queries and the modeling of the relationship component of the similarity score and **not** the use of the LLM as a parser. Our binding module is designed to learn a shared embedding space between open-ended object queries and visual parts. This allows us to extract visual slots that are relevant to the object query, enabling more accurate compositional understanding. The exact design choices are not straightforward. **We added extensive ablations about them in Appendix A.1**. Importantly, without the local contrastive loss that we propose, the relational understanding performance drops significantly as seen in the newly added Table 6\.
>
> **LLM-Based Parser as a Tool, not a Contribution**
>
> We want to emphasize that the specific LLM instantiation used in our parser is not a part of our contribution, but rather a tool that we utilize to obtain scene graphs. We consider an LLM-based parser as the best automated option currently available for this task.
>
> We **added a quantitative and qualitative ablation study (Appendix A.3)** that compares three different parser families:
>
> * LLM-based parsers (llama3)
> * Automatic parsing based on Spacy
> * Supervised scene graph parsers (T5-based)
>
> This ablation study provides valuable insights into the strengths and weaknesses of each parser family, illustrating some important failure modes **(see added Table 9 in the Appendix)** of automatic and supervised scene graph parsers that impacts downstream compositional performance **(see added Figure 6 in the Appendix)**.
>
> **Comparison with TagAlign**
>
> We were not aware of this work and thank the reviewer for pointing us to it. We added a comparison in the added Scene Graph Discussion section (Appendix A.3). TagAlign also uses an LLM-based text parser, but there are significant differences between both approaches :
>
> - **Finite vs open-ended objects**: TagAlign adds a fine-grained classification objective to the image representation by extracting a **finite set** of tags for objects and attributes. In contrast, our approach focuses on capturing **open-ended objects** and does not require the extraction of a finite set of tags. The fine-grained alignment is learned with a contrastive objective.
> - **Relationships**: Moreover, TagAlign does not handle relationships, which is a crucial aspect of our approach. Enumerating a finite set of triplets (predicate, object, subject) does not seem reasonable, as it would lead to an **explosion of possible combinations**.
>
> We additionally hypothesize that the classification objective in TagAlign might actually amplify the bag-of-words behavior when representing multiple objects. For example, "red apple and green banana" and "green apple and red banana" would be represented by the same classification labels measuring the presence of “red” and “green” attributes as well as “apple” and” banana” objects without any binding distinction. Unfortunately, the weights are not available, so we were unable to add this to the comparison.
>
> In conclusion, while TagAlign shares some similarities with our approach, there are significant differences in both methods. We believe that our approach provides a more comprehensive understanding of relationships between objects and is better suited for open-ended compositional reasoning tasks.
>
> **Parsing Cost**
>
> We performed the parsing by serving N instances of Llama3-8b on v100 machines. Each dataset is then chunked into K processes that do not require any GPUs and send requests to the served LLM parsers through vllm to maximize the throughput of the parallelized requests.
>
> For instance we parsed the COCO dataset (∼ 500k captions) parallelizing N=10 instances of the parser, with K=128 chunks in 3.5 hours. Similarly, we parsed the Visual Genome dataset (∼ 200k captions) with N=8 instances, K=64 chunks in 1.7 hours. We added the details in **Appendix A.3.**
>
> We hope that this analysis addresses the reviewer's concern and provides a clearer understanding of the computational cost associated with using an LLM for parsing
>
> \[1\] vllm: [https://github.com/vllm-project/vllm](https://github.com/vllm-project/vllm)

---

> > ### Author Response · Authors · 2024-11-22
> > **Rebuttal Answer - Part 2**
> >
> > **Can some MLLMs be regarded as text parser?**
> >
> > Yes, some MLLMs, such as InternVL and LLaVA, can be considered as a tool to obtain scene graphs. They could help **disambiguate the meanings** of some caption and make them more aligned to the actual visual content.
> > However, using MLLMs as text parsers comes with **additional computational costs** compared to simple LLMs. Moreover, when evaluating the performance of these models on benchmark datasets, there is a risk that the MLLM might correct negative candidate captions to match the image or add objects that were not initially mentioned in the caption based on the image content, thereby changing the taxonomy of some evaluation benchmarks.
> >
> > We hope our answers addressed the reviewer concerns and are happy to tackle any other questions.

---

> > > ### Comment · Reviewer_ar8U · 2024-11-26
> > > **Official Comment by Reviewer ar8U**
> > >
> > > I thank the authors for their response.All my concerns have been addressed. I will keep my original score of 6.

---

> > > > ### Author Response · Authors · 2024-11-26
> > > > **score increase clarification**
> > > >
> > > > We thank the reviewer for their answer. Could the reviewer point us to what additional information we can give to get an increased score given al the concerns have been addressed ?

---

> > > > > ### Author Response · Authors · 2024-11-28
> > > > >
> > > > > We thank the reviewer. We would love to hear from them about the potential missing components to recommend acceptance?

---

### Official Review · Reviewer_kw33 · 2024-11-04

**Soundness:** 3
**Presentation:** 3
**Contribution:** 3
**Rating:** 5
**Confidence:** 3

**Summary:**

The paper introduces OC-CLIP, a novel approach aimed at enhancing the compositional understanding capabilities of CLIP-like models.
The key innovation is the integration of a binding module that connects a scene graph derived from text descriptions with a slot-structured image representation.
The proposed structured similarity score between the two modalities seems to capture interactions between objects and their contextual relationships more effectively.

**Strengths:**

(1) The introduction of the binding module is well-motivated and technically sound.
(2) The structured similarity score, composed of an object scoring function and a relationship scoring function, seems a reasonable method for modeling the scene graph and the visual slots relationships.

**Weaknesses:**

(1) The binding module and structured similarity score introduce additional complexity to the CLIP model. The authors should provide an analysis of the trade-offs of computational overhead and performance gains.

(2) The authors evaluate OC-CLIP on several standard benchmarks of compositional understanding and spatial relationship understanding. However, as a fundamental multi-modal representation model, additional downstream experiments on tasks such as zero-shot image classification, video action recognition, geo-localization, text-to-image retrieval etc. should strengthen the paper.

**Questions:**

see weaknesses

---

> ### Author Response · Authors · 2024-11-22
> **Rebuttal Answer**
>
> We thank the reviewer for their feedback, it allowed us to improve our work considerably. In the following we tackle the raised points.
>
> We have made those following additions based on the reviewers concerns and detail them further below :
>  * we trained both OC-CLIP and CLIP from scratch on CC12M and compared their zero-shot classification and compositional understanding.
>  * we added a computational analysis of our binding module (in terms of FLOPs) as suggested.
>
> **Additional Downstream Tasks**
>
> During the rebuttal period, we scaled up our experiments to train both CLIP and OC-CLIP **from scratch on CC12M dataset** to demonstrate the scaling potential of our approach. We train both models from scratch on 20 epochs of CC12M and report zero-shot downstream classification performance on the ELEVATER [1] suite in the newly added Table 7 and Table 4\. Our results show that OC-CLIP achieves significant performance gains in **zero-shot classification (e.g. \+9.2% in ImageNet)** compared to CLIP while maintaining a significant gap in compositional understanding  in the **zero-shot sugarcrepe swap attribute (+15.9%) and swap obj (+14.3%) splits**.
>
> We hope that these additional results address the reviewer's concern on the additional downstream performance of OC-CLIP and the usefulness of the introduced **inductive biases to mitigate the bag-of-word behaviour** previously identified in CLIP-like models.
>
> **Computational analysis**
>
> We also addressed the concerns regarding the computational overhead introduced by the binding module in OC-CLIP in the **Appendix Table 8**. We acknowledge that the binding module introduces additional complexity to the CLIP model, which can result in increased computational overhead due to the cross-modal attention operations. But, unlike CLIP’s text encoder, OC-CLIP only needs to encode information about single open-ended objects and relationships and thus requires much less capacity than CLIP that needs to encode whole sentences composed of multiple objects and relations between them. With this observation, we employed several strategies the in the experiments that we added on CC12M **(Section A.2 of the Appendix)**:
>
> * We used a smaller embedding size (256 vs 512\) and number of layers (6 vs 12\) in the text encoder.
> * We operated on a reduced embedding space (256) for the binding module and projected the ViT-B-16 patches from a 768 to a 256 embedding space before computing the nodes to patch cross-attention logits.
>
> These optimizations allowed us to reduce the computational overhead while maintaining the performance gains achieved by our approach.
>
> To additionally quantify the resulting computational overhead, we performed an analysis of the binding module's impact on the overall computational cost of OC-CLIP in the added Table 8\. Our results show that the binding module introduces an overhead when using a base architecture (2.2x FLOPS), but this overhead is reduced when scaling the ViT backbone to a L backbone **(1.3x FLOPS)** because the cost of the binding module is not the bottleneck anymore.
>
> We hope that this analysis addresses the reviewer's concern and provides a clearer understanding of the trade-offs between computational overhead and performance gains in our approach. We are happy to answer any further questions the reviewer may have.
>
> [1] ELEVATER : ELEVATER : https://computer-vision-in-the-wild.github.io/ELEVATER/

---

> > ### Author Response · Authors · 2024-11-26
> > **Rebuttal answer request**
> >
> > We thank the reviewer for their time. We think we addressed all the concerns of the reviewer and are looking forward to hearing from them.

---

> > > ### Author Response · Authors · 2024-11-28
> > >
> > > We thank the reviewer again and would love to hear from them regarding our rebuttal answer ?

---

### Author Response · Authors · 2024-11-22
**General Rebuttal Answer - Part 1**

We thank all the reviewers for taking the time to read and review our paper. We believe these reviews helped us **considerably improve our work**. We answer each review separately but this comment is to address and comment on some recurring points of the majority of the reviewers.

We would first thank all the reviewers for acknowledging our work to be well-motivated by the importance of improving compositional reasoning (Rev. **kw33** and **uY5M**) and our newly proposed binding module to be simple and reasonable yet novel, interesting, and sound (Rev. **kw33**, **ar8U**, **uY5M**, **HgdJ**). Moreover, we are glad to see that Rev. **ar8U**, **uY5M**, **HgdJ** highlighted the empirical effectiveness of our method.

Importantly we made the **following main changes** in the paper  ( and highlighted them in blue in our draft) :

- We trained CLIP and OC-CLIP from scratch on CC12M in addition to training the OC-CLIP’s binding module on top of a pre-trained CLIP on the COCO dataset. \[Addressing concerns of Reviewers **kw33**,**uY5M** and **HgdJ** about general understanding capacity of OC-CLIP\]. And show performance gains of OC-CLIP (**\+9.2% on ImageNet**).
- We included results on additional benchmarks: ARO/VL-Checklists and the ELEVATER datasets \[As asked by Reviewer  **HgdJ**\], and show our method is competitive in all of them
- We included new baselines: DAC, DCI and CLIP-SVLC \[As asked by Reviewers  **HgdJ, uY5M** \] and show that they still fall behind OC-CLIP in splits requiring precise attribute-binding despite their large-scale finetuning.
- We introduced additional ablations and explored different scene parsers, architecture components and loss terms. \[ As suggested  by Reviewers **HgdJ,uY5M, ar8u**  \] and show the importance of OC-CLIP design choices.
- We presented a computational analysis of OC-CLIP \[ As asked by Reviewer **kw33**\], evaluating computational overhead vs performance gains.


We give further details below :

**Additional compositional Baselines/Experiments**

1. Added Baselines

In our study, we focused on comparing OC-CLIP to  hard-negative-based baselines that generate hard negatives **in-distribution**, specifically from the COCO/VG/Flickr domain. This is why we included methods such as NegCLIP, CE-CLIP, and CounterCurate, which also focus on generating hard negatives within these specific domains.

However, we acknowledge the importance of broader comparisons and have now included additional methods such as DAC,  DCI and CLIP-SVLC  to our analysis (see Table 2 and Table 5).. These methods are different in that they incorporate both hard negative mining on a **larger scale** and **dense captioning/recaptioning**, which is not the primary focus of our work. Nonetheless, their inclusion provides a more comprehensive comparison and allows us to better position our approach within the broader landscape of compositional reasoning benchmarks.

Notably, we see that on the sugarcrepe/ARO splits that assess for precise object-attribute binding (swap attribute/ARO-A) all these methods still fall behind :

|  | OC-CLIP | DAC-LLM | DAC-SAM | DCI | CLIP-SVLC |
| :---- | :---- | :---- | :---- | :---- | :---- |
| Swap-Att | 88.5 | 74.1 | 75.3 | \\- | \\- |
| ARO-A | 82.0 | 73.9 | 70.5 | 67.6 | 73.0 |

Which confirms the results we presented with the synthetic experiments that the hard-negative approach, even at scale,  is not sufficient to solve the bag-of-words behavior of CLIP and justifies the inclusion of more structured inductive biases within the architecture of the model like we do for OC-CLIP

2. Added Benchmarks

Our goal is to evaluate CLIP-like models compositional understanding in **plausible and grammatically correct cases**, the SugarCrepe benchmark  has identified exploitable  textual biases in previous mainstream procedurally-generated hard negatives benchmarks like the COCO and Flickr set of ARO and VL-checklist. Specifically the \[1\] shows that procedurally generated hard negatives are either highly grammatically incorrect and can be identified by a blind model or by a good language model. SugarCrepe proposes to follow the same fine-grained taxonomy on attributes, objects, relationships  as VL-checklist but ensures that the hard-negative are not distinguishable by a blind model. However, as requested by the reviewer we **added the results on the full ARO suite and VL-Checklist  in Table 5**. We also added DAC, DCI, SVLC as suggested as baselines in sugarcrepe and ARO/VL-checklist.

---

> ### Author Response · Authors · 2024-11-22
> **General Rebuttal Answer - Part 2**
>
> **Zero-shot Classification**
>
> Several reviewers had concerns regarding whether OC-CLIP retains its original image understanding capabilities after fine-tuning with the scene graph.  In our compositional understanding experiments, we compare our approach with data-centric fine-tuning methods that do not add any additional parameters. These methods are expected to retain some of the general capabilities of the initial backbone. However, our binding and relationship modules are trained **from scratch on a smaller scale dataset**, which means they may not generalize as well to unseen data and can only be expected to work well within the vocabulary domain they have been exposed to (e.g., COCO/VG/GQA in our experiment setting).
>
> In order to assess how training on scene-graph affects general understanding capabilities, we **added experiments where both CLIP and OC-CLIP architectures are trained from scratch** on a noisy and non-human curated dataset , **CC12M** \[1\]. This allows us to compare both models' general understanding and compositional downstream performance in the newly added Table 4 and 7\. Notably we notice an improvement  in general understanding with **\+9.2% on Imagenet zero-shot classification** performance while maintaining a significant gap in zero-shot sugarcrepe s**wap attribute (+15.9%) and swap obj (+14.3%) splits**. We added the extensive **zero-shot classification results on the ELEVATER \[2\] suite in Table 7** and we leave further scaling experiments for future work.
>
> **Computational Analysis**
>
> Some reviewers asked to comment on the trade-offs of the computational overhead and performance gains of our method.  We acknowledge that the binding module introduces additional complexity to the CLIP model, which can result in increased computational overhead due to the cross-modal attention operations. However we were able to mitigate it in the scaling experiments because of the way OC-CLIP binds objects to visual parts. Importantly, unlike CLIP’s text encoder, OC-CLIP only needs to encode information about single open-ended objects and relationships and thus requires much less capacity than CLIP that needs to encode whole sentences composed of multiple objects and relations between them. With this observation, we employed several strategies the in the scaling experiments that we added on CC12M **(Section A.2 of the Appendix)**. We added in **Table 8** a quantitative analysis of the FLOPS comparison of CLIP and OC-CLIP and show that our binding module is not a bottleneck any more when scaling the ViT backbone.
>
> We hope that this analysis addresses the reviewer's concern and provides a clearer understanding of the trade-offs between computational overhead and performance gains in our approach.
>
> **Scene Graph Discussion**
>
> Finally we want to stress the fact that the specific LLM instantiation is not a claimed contribution but rather a tool used in the proposed method.  In particular, we consider an LLM-based parser as the best automated option currently available for obtaining scene graphs.
>
> To support this claim, we have **added a quantitative and qualitative ablation study** (Appendix A.3) that compares three different parser families:
>
> * LLM-based parsers (llama3)
> * Automatic parsing based on Spacy
> * Supervised scene graph parsers (T5-based)
>
> Most notably we notice **in Figure 5** the strength of the parser impacts the relational understanding performance with a swap obj (llm) : 80.7 (supervised) :77.5, (automatic) : 71.4 and replace rel (llm) 80.6, (supervised) 74.3, automatic (66.5) which is consistent with the **qualitative failure modes identified in Table 9\.**

---

### Meta-Review · Area_Chair_9MtG · 2024-12-18

**Metareview:**

This paper focuses on improving the compositional understanding of pre-trained CLIP-like models without using additional data annotations. To achieve this, the paper introduces a binding module relying on a scene graph of the text to facilitate structured similarity assessment. The idea of the paper is well-explained and discussed with ablations/experiments/comparisons. The introduced binding module might incur extra costs for inference, and due to the nature of the design, the original capabilities of the CLIP model might be lost. Moreover, the paper still lacks enough discussion about the direct relationship between the quality of the compositional data (either generated by LLMs or other tools) and the final evaluation. Based on these weaknesses pointed out by the reviewers, I would not recommend accepting the paper.

**Additional Comments On Reviewer Discussion:**

- Reviewers kw33 and uY5M asked about missing evaluations and baselines. The authors added several zero-shot evaluation numbers and several retrieval evals of hard negative mining methods. It seems that not all the downstream tasks are evaluated, which should be fine given the limited rebuttal period.
- Reviewer ar8U question about the runtime and the necessity of using LLMs for scene parsing. The authors provide a list of other approaches. However, not all the popular LLMs are included in the study. However, it seems the reviewer is fine with the provided rebuttal.
- Reviewer uY5M pointed out the absence of a detailed study of each component, in particular, the loss component. I think the rebuttal from the authors is still not as detailed as the reviewer requested. Moreover, the authors show that the other capabilities of CLIP are indeed lost
- Reviewer HgdJ raised concerns about the scene graph composition and the measurement of its quality. It seems the paper still needs some work to justify the use of scene graphs and how to improve it for certain datasets.

---

### Decision · Program_Chairs · 2025-01-22

Reject